# The honesty behind tears: Situational, individual, and cultural influences on the perception of emotional tears as sincere

Monika Wróbel[1*], Janis H. Zickfeld[2], Paweł Ciesielski[3]

1 Institute of Psychology, University of Lodz, Lodz, Poland, 2 Department of Management, Aarhus University, Aarhus, Denmark, 3 Doctoral School of Social Sciences, Institute of Psychology, Adam Mickiewicz University, Poznan, Poland

* monika.wrobel@uni.lodz.pl

## Abstract

Emotional tears have been considered honest and sincere signals, most likely because they are difficult to shed on demand. At the same time, people acknowledge that tears can be strategically used to manipulate others – so-called *crocodile tears*. Hence, the question arises under what circumstances tears are perceived as honest signals and when as crocodile tears. Here, we investigate this question across three experimental studies and diverse populations. In a preliminary study ($N = 7{,}007$), we obtain the first evidence that emotional tears can increase perceptions of honesty, which might vary according to the situational context or the gender of the target. In two main studies ($N = 3{,}488$) using a varied pool of standardized and non-standardized portraits of tearful and non-tearful targets presented in different potentially manipulative vs. non-manipulative contexts and varied in their warmth, we test perceptions of honesty across five countries (Norway, Poland, South Africa, Canada, and the UK). Overall, the main effects are weak and suggest that perceptions of honesty depend on target characteristics, situational factors, and observer characteristics. We observe some evidence that emotional tears increase perceptions of honesty more strongly for targets low in warmth (experimentally manipulated via facial features or via target gender), which also affects support intentions. Manipulative contexts slightly reduced perceptions of honesty, but these effects were moderated by target characteristics. Individuals scoring high on psychopathy showed lower ratings of honesty for targets with emotional tears. Together, these findings provide further evidence that whether emotional tears signal honesty likely depends on various individual, situational, and cultural factors. The small effect sizes call for improved manipulations and more ecologically valid designs in the future.

**Data availability statement:** Data, materials (except stimuli), analysis syntax, and preregistration are available at https://osf.io/v5fgy/.

**Funding:** The research was funded by the National Science Centre, Poland (https://www.ncn.gov.pl/en) by a grant awarded to Monika Wróbel (Grant Number: 2021/43/B/HS6/01882). The funders did not play any role in the conceptualization or preparation of the studies reported in this manuscript.

**Competing interests:** The authors have declared that no competing interests exist.

## Introduction

In 2008, during the Democratic primary, Hillary Clinton teared up at a campaign event in New Hampshire. Her mild emotional outburst was heavily featured across all media for the next few days. Many people accused her of faking the tears to gain final votes for the upcoming election, while others praised her weeping as a rare and welcomed expression of emotionality [1]. It is intriguing to see that an emotional expression such as tears can be interpreted in dramatically different ways, and the question opens up in what circumstances people believe that shedding tears is a sign of honesty or the polar opposite.

On the one hand, some evidence indicates that human emotional tears are positively associated with perceived honesty [2,3], reliability [4], sincerity [5,6], and trustworthiness [7]. This positive effect of tears on honesty-related ratings most likely stems from the fact that producing tears on cue is exceedingly difficult. Similar to other non-verbal signals that cannot be voluntarily controlled (e.g., blushing, pupil dilation, goosebumps, sweating [8,9]), tears are considered inherently honest, direct indicators of true internal states (for instance, emotions expressed by virtual humans become more believable when accompanied by involuntary non-verbal signals, including tears [10]). Accordingly, tears are typically regarded as a genuine social signal [6,11–14] communicating that the crier is overwhelmed and needs help [3,15,16]. A typical response to this signal, as recently proposed by the social glue model of emotional tears [14], is compassion and increased willingness to support the tearful individual. Hence, shedding tears may be socially beneficial because of their influence on others.

On the other hand, as illustrated in the example above, the positive connotations of tears are often challenged by conventional wisdom, which underscores tears' usefulness in manipulating others and portrays tearful individuals as dishonest. For instance, a Google search yields over 20 million hits for the phrase "tears and manipulation," with many magazine articles and online forums revolving around fake tears. Popular literature also lists shedding tears as a widely used manipulation tactic [17,18] and qualifies "manipulative tears" as a separate category of crying episodes [19]. This suggests that in some circumstances, tears may be seen as a calculated behavior used to exert *social influence* (e.g., deliberately evoke compassion in the observer or make them provide help). In such circumstances, shedding tears might paradoxically alert the observer even more, suggesting that tears are shed only to create a "smokescreen", the aim of which is to conceal the crier's malicious intentions and render them a person who can be trusted. However, scientific evidence regarding tears used for manipulation is missing.

Here, we take a closer look at the paradox between regarding tears as a sincere social signal and their manipulative reputation, by analyzing the conditions under which tearful individuals are perceived as honest or not. We test mediating factors of why people in tears might be perceived as honest or not, and how such perceptions affect the observer's support intentions. Drawing on the idea that the social signal value of emotional expressions is shaped by the context in which these expressions appear [20], we focus on contextual factors that may influence the perceived authenticity of tears and the perceived manipulativeness of the crier.

**The perceived honesty of a tearful individual**

The manipulative reputation of tears is reflected by the popular phrase "crocodile tears", which derives from a myth that crocodiles shed tears while eating their prey [21]. The idea behind this phrase is that people shed tears deliberately for malicious reasons to make a fake impression that they are overwhelmed by sadness, grief, or remorse. Such tears differ from spontaneous, sincere emotional tears because they are shed on demand to influence others [12,22–25]. Crocodile tears differ from genuine tears in terms of social function rather than the objective indicators of authenticity [12]. For instance, when people fake tears by recalling the memories that made them cry in the past, the tears per se are "real," but they are considered inauthentic because they do not flow naturally to express emotions related to the memory but are shed deliberately to exert influence on others. In consequence, genuine and fake tears may look exactly the same, which makes them difficult, if not impossible, to discern based on the mere objective characteristics. This most likely explains why people are often emotionally moved by movies with actors faking tears or why they are motivated to provide help regardless of whether they see pictures of individuals with artificial, digitally added tears [e.g., 3, 26] or pictures of individuals crying spontaneously [e.g., 27, 28]. Yet, even though there are no objective standards for distinguishing between fake and genuine tears, and people are highly inaccurate at differentiating between the two, they are biased by their subjective perception of tears' authenticity. For instance, Van Roeyen et al. [24] showed that individuals whose tears were subjectively evaluated as inauthentic were perceived as less reliable, less warm, more manipulative, and less suitable for trustworthy professions (e.g., teachers), even when their tears were, in fact, genuine. Accordingly, the perceived authenticity of the target's expression (rather than the objective characteristics of tears) seems to be a crucial factor accounting for the target's perceived honesty.

It is also likely that when rating the honesty of a tearful individual, observers may rely on their judgments about this person's manipulativeness. Such judgments, similar to perceptions of expression authenticity and other social judgments [29], are subjective and based on more or less vague cues (e.g., "Is the crier a person I can trust?," "Is there anything suspicious in the situation that would make the crier fake tears?," "Can people, in general, be trusted?"). Accordingly, the perceived honesty of a tearful individual may be shaped by the overall subjective evaluation of the authenticity of tears as well as the tearful person's manipulativeness in the specific context in which tears appear. Such contextual factors, as we argue below, may include the features of the *situation*, the *target*, the *observer*, and the *culture*.

**Situational context and target characteristics**

Much empirical evidence indicates that the social meaning of emotional expressions is deeply embedded in social context (for reviews, see [20,30]). For instance, although a smile is an inherently affiliative social signal, its meaning changes to non-affiliative when it is shown on the face of a rival in a competitive context [20]. Perception of tears also changes depending on contextual factors. For instance, shedding tears in situations that do not provide any clear reasons for tearing up decreases the target's perceived competence, but this effect is not observed for tears shed in emotional situations [31]. Similarly, tearing up in work settings, in contrast to relationship settings, is regarded as less appropriate and reduces the crier's competence, especially for men [32].

It seems likely that the perceived authenticity of tears, as well as the perceived manipulativeness of a tearful target, are also modulated by situational context. Anecdotal evidence suggests that tears should be regarded as less authentic, and tearful individuals should be seen as more manipulative in settings that promote manipulative behaviors, such as courtroom or political contexts [12,25]. An example of this common belief is the already-mentioned case of Hillary Clinton [1,33] or a ban issued in 2008 by state prosecutors in Ohio, which prohibited defense attorneys from crying on cue in death penalty cases because such crying was considered emotional blackmail [34]. Examples like these imply that situational contexts in which people may benefit from exerting social influence on others (e.g., making the impression that they feel sadness, regret, or remorse; evoking compassion in others; or presenting themselves as credible) seem particularly likely to decrease perceived expression authenticity and increase perceived target manipulativeness.

The social signal value of emotional tears may also depend on the characteristics of the target, because these characteristics may be treated as indicative of the target's good vs. bad intentions. For instance, high perceived warmth informs the observer that the target's intentions are pure and friendly, while low perceived warmth is a signal of dishonest, malicious intentions [35,36]. Accordingly, people perceived as low in warmth may be seen as more predisposed to shed crocodile tears than those high in warmth, which aligns with the fact that warmth and honesty are positively correlated [37–39]. Importantly, inferences about people's warmth can be made on the basis of their physical (in particular, facial) appearance, because people evaluate faces on various social and personality dimensions [40–42]. For instance, prominent cheekbones and high inner eyebrows are associated with higher warmth, while flat cheekbones and low inner eyebrows are perceived as signals of low warmth [40]. Consequently, some people may have warmer-looking faces than others and, hence, be assessed as warmer in general [40,43]. As such, facial features may guide social interactions, meaning that when there is no other information about the target provided, observers may evaluate the target's dispositions and intentions based on their facial features. This may, in turn, affect the perceived authenticity of the target's expression and their perceived manipulative tendencies.

Drawing on the idea that tears are generally considered a sincere emotional signal that is positively linked to perceived honesty, but these perceptions are likely to change due to the manipulativeness of the situational context and the warmth of the target's face, we formulated two hypotheses regarding the effect of tears on perceived honesty:

H1:  Tearful individuals will be perceived as more honest than non-tearful individuals, but this effect will be moderated by the warmth of the target face and the manipulativeness of the situational context. The effect of tears on honesty will be stronger for high-warmth faces (compared to neutral and low-warmth faces). The effect of tears on honesty will be reversed for manipulative situations, so that showing tears in manipulative (vs. non-manipulative) situations will decrease perceived honesty.

H2:  The effect of tears on honesty will be mediated by the perceived authenticity of the target's expression and the manipulativeness of the target. This will be further moderated by the warmth of the target face and the manipulativeness of the situational context. Tears will increase perceived expression authenticity and decrease perceived target manipulativeness for non-manipulative situations and high-warmth faces, while the effect will be reversed for manipulative situations and smaller for low-warmth faces. Perceived expression authenticity will be positively associated with perceived honesty, while the target's perceived manipulativeness will be negatively associated with perceived honesty.

Finally, based on the idea that the positive effects of tears on support intentions may be driven by the fact that tears are considered sincere signals that render the target an honest person [4,11,16], we also predicted that tears would be positively related to support intentions, and perceived honesty would account for this effect:

H3:  Tearful individuals will evoke stronger support intentions than non-tearful individuals, and this effect will be mediated by perceived honesty. Tearful faces will be perceived as more honest, which will be associated with increased support intentions. This effect will be moderated by the warmth of the target face and the manipulativeness of the situational context (see H1).

Note that physical characteristics of the face may also signal competence (the second of the "Big Two" social dimensions [44]). However, we did not formulate any hypotheses about the effects of the target's face competence because this dimension plays a secondary role in judging the target's intentions [35,36,45], and hence, its relationship to perceived honesty is not straightforward.

## Observer characteristics

Given that people, in general, vary in the tendency to regard others as truthful [46], some observers may be more likely than others to undermine the authenticity of tears and evaluate the target as manipulative and dishonest. This should

refer to those who are suspicious of other people's intentions. For instance, personality researchers have established that the Dark Triad personality traits are associated with negative perceptions of others, low empathy, and generalized social distrust [47–50]. In line with these findings, Dawel et al. [22] found that the affective features of psychopathy (e.g., callousness and low empathy) are related to a reduced tendency to appreciate the authenticity of others' distress. This evidence suggests that individuals characterized by dark aspects of personality may perceive others' tears as inauthentic and ascribe negative characteristics to tearful targets. Based on this reasoning, we expected that the observer's Dark Triad personality traits would further moderate the effect of tears on perceived honesty:

H4: The occurrence of tears, situational context, and the observer's dark personality traits will interact when affecting perceived honesty. The positive effect of tears on perceived honesty in non-manipulative situations will be stronger for individuals scoring low on dark personality traits. The negative effect of tears on perceived honesty in manipulative situations will be stronger for individuals high on dark personality traits.

The overview of our hypotheses is presented in Fig 1.

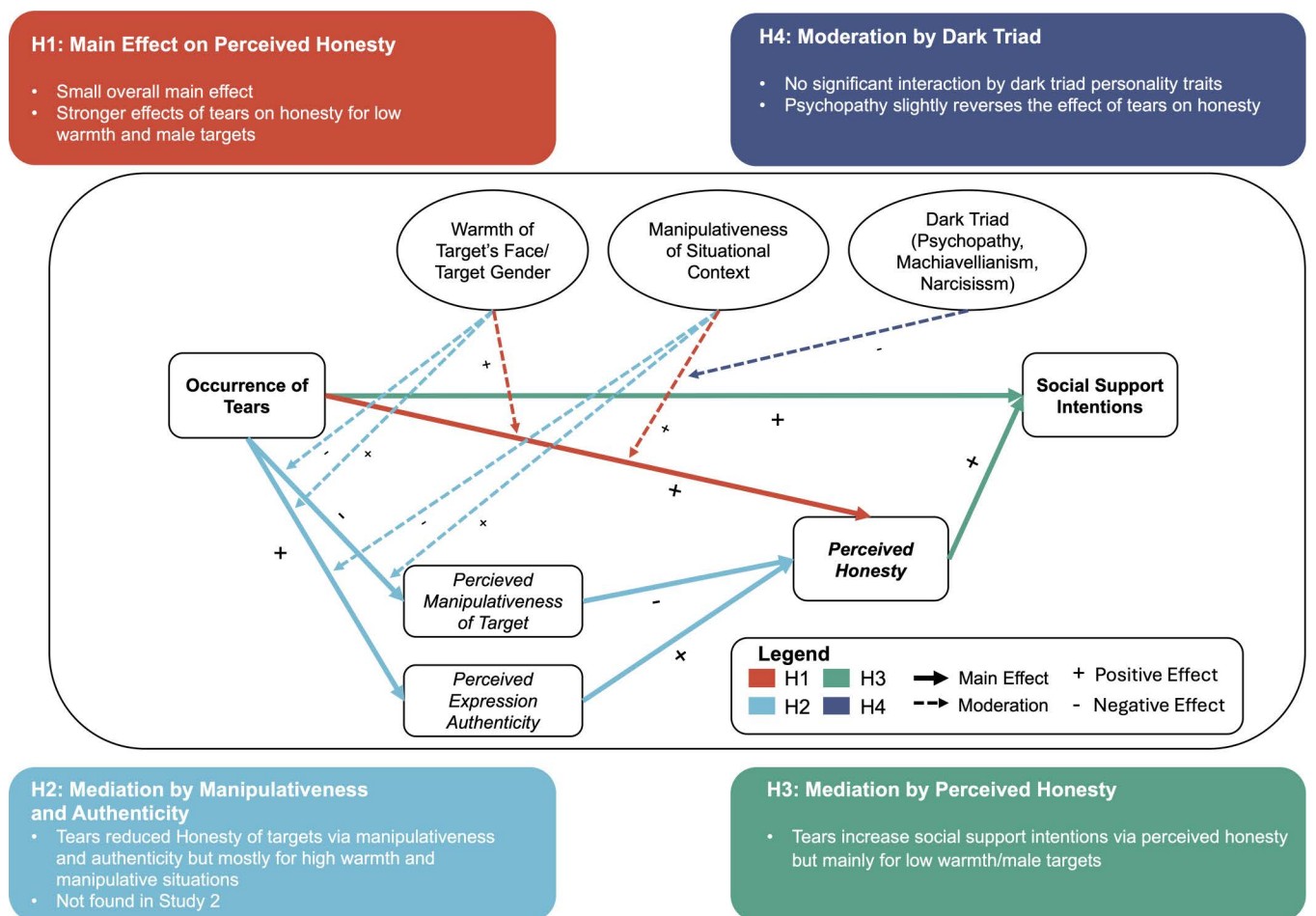

**Fig 1. Overview of Hypothesized Relationships and Main Findings.** Arrows depict hypothesized relationships; boxes summarize the main findings.

## Other variables influencing the perceived honesty of tearful targets

Apart from the variables mentioned above, there are other factors that may potentially influence the perceptions of a tearful individual's honesty. Yet, the current evidence is insufficient to formulate specific hypotheses regarding the role of these factors.

For instance, conventional wisdom holds that the tendency to shed crocodile tears may be more typical of women than men [12]. A study by Buss et al. [51] vaguely supports this reasoning by showing that women are more likely than men to use regression tactics (i.e., crying, whining, and pouting) to achieve their goals in close relationships. At the same time, women, in comparison to men, are typically perceived as warmer [36,52,53], which aligns with the fact that facial features indicative of warmth are more pronounced in women than men [54]. Some studies also show that they are generally less likely to engage in emotional manipulation [55], which suggests that they should also be less likely to shed tears for manipulative purposes. These observations point to the possible differences in the overall perception of tearful men and women, which is supported by a recent study that has shown that women crying for manipulative purposes evoked more anger and less empathy than men crying for manipulative purposes [56]. However, this study was published after we pre-registered our hypotheses and conducted Study 1.

Studies also show that the perception of tears and their social meaning is moderated by the perceived appropriateness of the expression. For instance, testing the social effect of tears in 41 countries revealed that tears evaluated as less appropriate were also negatively linked to perceived warmth and helplessness, thereby reducing participants' social support intentions [3]. Yet, the same research also showed that perceived appropriateness strongly depended on whether tears matched the context in which they appeared (i.e., tears shed in emotional situations were regarded as more appropriate than tears shed in neutral situations). This suggests that manipulative context may potentially change perceptions of expression appropriateness, but the exact nature of this relationship is difficult to predict because it is unknown whether shedding tears in a manipulative social context tends to be perceived as matching this context or not.

Finally, the expression and perception of tears also vary across different cultures [57]. A recent cross-cultural project revealed that tears affected support intentions differently across 41 countries [3], which was possibly driven by culture-specific social or gender norms that regulate the expressions of crying [57] or cultural differences in trust [58–60]. These findings indicate that cultural factors may be important when analyzing the effects of tears on perceived expression authenticity, as well as perceived target manipulativeness and honesty, but we did not have any specific hypotheses regarding this issue.

## The present research

The current research was motivated by the observation that emotional tears, despite being difficult to fake and thus generally perceived as sincere, are paradoxically considered a means of manipulation and a signal of dishonesty. Driven by the idea that this ambiguity may stem from the fact that the subjective perception of tears changes depending on the conditions in which tears appear, we investigated how these conditions may affect the perceptions of honesty of tearful individuals, and how these perceptions further influence the observer's support intentions. To address these questions, we conducted three studies. First, we ran a Preliminary Study re-analyzing data collected by the Cross-Cultural Tears (CCT) project [3] across 7,007 participants and 41 countries. This project included a measure of perceived honesty, but no results regarding this measure had been published previously [3,31]. Here, we explored the role of social context for perceived honesty and the interaction with perceived appropriateness and target gender. As the study was not designed to test our hypotheses and its nature was mostly exploratory, we were limited by the specific measures collected. Second, we performed Study 1 across four countries ($N = 1,893$), presenting participants with tearful vs. non-tearful targets and manipulating contextual factors such as the warmth of the target's face and the manipulativeness of the situation in which this target was presented. We also tested the mechanisms behind the effect of tears on perceived honesty, focusing on the perceived authenticity of the target's expression and the perceived manipulativeness of the target. In addition, we

examined the moderating role of the observer's dark personality traits for inferences of honesty and performed exploratory analyses addressing the role of target gender, perceived appropriateness, and culture. Finally, we conducted Study 2 ($N$ =1,595) to replicate and extend the results of Study 1 in a different sample, using more ecologically valid stimuli and a modified manipulation of perceived warmth.

## Open practices

For the re-analysis of the CCT project, ethical approval was granted at each participating institution [3]. Analyses were not pre-registered and are exploratory. All data, materials, and analysis syntax are available at https://osf.io/fj9bd/. Tearful pictures are available at https://www.chicagofaces.org/resources/.

For the two main studies, the research received approval from the Ethics Committee of the University of Lodz (1/KEBN-UŁ/I/2022–23), and all participants provided written informed consent to participate in the studies and for the publication of their anonymized data. Our design, hypotheses, and analysis plan were registered prior to the start of data collection, and deviations from the pre-registered protocol are explicitly mentioned (Study 1: https://osf.io/k4ev9; Study 2: https://osf.io/p7hxe). In addition to the pre-registered analyses, we run a series of exploratory analyses addressing the role of other moderating variables. We report all exclusions, manipulations, and measures. Data, materials (except stimuli), analysis syntax, and preregistration are available at https://osf.io/v5fgy/. Pictures used in Studies 1 and 2 are not available online due to copyright restrictions and can be requested by contacting the first author. The individual in this manuscript (shown in Fig 3) has given written informed consent (as outlined in PLOS consent form) to publish their portraits during the initial validation of the dataset this image was taken from [43].

## Preliminary study: Re-analysis of the CCT project

As a first step, we re-analyzed openly available data collected from 7,007 individuals across 41 countries by the CCT project [3], focusing on perceived honesty. Detailed information on the dataset and analyses is presented in the Supplementary S1 Note. With the dataset, we were able to partly test H1 and H3, but importantly, the dataset had been originally intended for a different research question, and hence, our analyses were exploratory. Nevertheless, the dataset provided a starting point as it includes ratings of perceived honesty provided by a large number of participants from 41 countries worldwide.

An overview of the main findings is provided in Fig 2 and Supplementary S1 Note. The re-analysis corroborated the idea that emotional tears can increase perceptions of honesty ($d$=.28, 95% CI [.23, .33], $p$<.001), although with variation across different countries ($Q(40)$ = 127.03, $p$ < .001, $I2$=70.73 [56.37, 83.76]; Fig 2a). This points to the potential role of cultural moderators of the effect of tears on perceived honesty. In addition, we obtained the first evidence that the effect of tears on perceived honesty is moderated by social context. While we were not able to test the role of manipulative vs. non-manipulative contexts, we found that tears did not increase ratings of honesty for neutral situations ($d$=−.04 [−.09, .005]) in which tears were considered less appropriate and, although not measured, probably also as less authentic (Fig 2b). Ratings of honesty increased for tearful targets in negative ($d$=.58 [.53, .63]) and positive contexts ($d$=.57 [.53, .62]). We also obtained some evidence that the effects of tears might differ depending on the target gender, with stronger effects for male targets ($d$=.43 [.39, .46]) compared to female targets ($d$=.32 [.28, .36]) as males were rated as less honest at baseline (Fig 2c). Perceived appropriateness did not moderate the effect of tears on honesty (Fig S1). Finally, we observed evidence for a partial mediation of the effect of tears on support intentions by perceived honesty (indirect effect: $β$=.06 [.05, .07], Fig 2d).

## Study 1: Systematic investigation across four countries

To test our hypotheses, we designed an experiment investigating perceptions of honesty across a range of situational and individual factors. Following the results of the CCT re-analysis, which suggested that the effect of tears on perceived

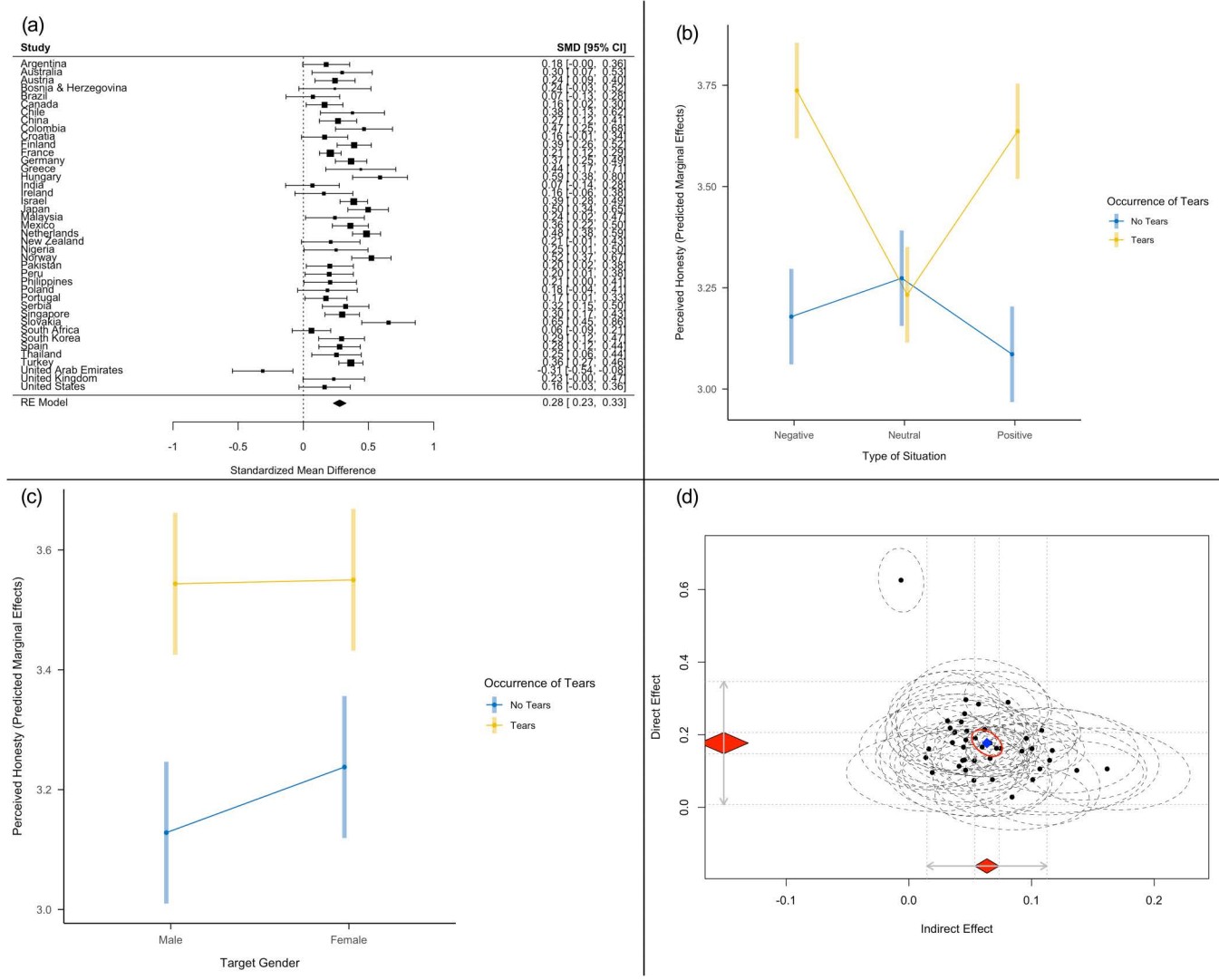

**Fig 2. Overview of Results of the CCT Project Re-Analysis: (a) Forest Plot of The Effect of Occurrence of Tears on Perceived Honesty across 41 Countries; (b) Interaction between Occurrence of Tears and Situational Valence on Perceived Honesty; (c) Interaction between Occurrence of Tears and Target Gender on Perceived Honesty; (d) Overview of Mediation Results of Occurrence of Tears on Support Intentions Via Perceived Honesty. (a, b, c)** Error bars specify 95% confidence intervals; **(d)** The X-axis specifies the indirect effect via perceived honesty, and the y-axis specifies the direct effect of the occurrence of tears on intentions-to-support when taking the mediator into account. **(d)** Red diamonds specify 95% confidence intervals, and gray arrows specify prediction intervals. Individual effects across the 41 countries and their confidence ellipses are plotted.

honesty might be moderated by cultural factors, we focused on four countries (Canada, Norway, Poland, and South Africa). These countries were selected based on two indices: (1) trust levels as measured by the World Value Survey [58] and the South African Social Attitudes Survey [61], and (2) perceived honesty effect sizes from the Preliminary Study (Fig 2a). Canada and Norway showed higher trust levels, while Poland and South Africa showed lower trust levels. Moreover, the effects of tears on perceived honesty also varied across the four countries, being the smallest for South Africa and the strongest for Norway (for details, see Supplementary S2 Note). Moreover, to expand on the finding that the effect of tears varies across different contexts, we went beyond negative, positive, and neutral situations and presented the targets in manipulative and non-manipulative contexts (differing in whether the target was presented as a person exerting social

influence). We also tested the mediation of the effect of tears on support intentions by perceived honesty and further explored the role of the target gender and perceived appropriateness.

## Method

**Participants.** We recruited a total of 2,014 participants from Canada, Norway, Poland, and South Africa. South African, Canadian, and Polish participants were recruited via Prolific.co and paid £1.7 for participation. Norwegian participants were recruited via Toluna and paid in Toluna points. Our sample size was determined based on the overall effect of tears on perceived honesty ($d = .28$) found in the Preliminary Study. Considering the aforementioned variations in this effect, we set our smallest effect size of interest (SESOI) to $d = .20$. Simulating a multilevel model in *simr* and using random effect structures based on Zickfeld et al. (2021), suggested a power of 80% at a sample size of around 450 per country. After excluding participants who a) completed the survey faster than ⅓ of the median time, b) were younger than 18 years of age, c) failed the attention check, and d) failed the comprehension check for all their ratings, the final sample consisted of 1,893 participants (926 women, 931 men, 26 non-binary, 6 other) ranging from 18 to 85 years of age ($M = 34.1$, $SD = 13.6$; see Table 1 for a detailed overview per country). The final dataset included 9,299 individual observations.

**Procedure.** We employed a 2 occurrence of tears (tears vs. no tears) x 5 target face (neutral, high warmth, low warmth, high competence, low competence) x 2 situational context (non-manipulative vs. manipulative situations) x 2 target gender (male vs. female) mixed design. We did not focus on variations in competence in the present study, but included it for the sake of completeness and exploratory reasons (a brief overview of the findings is provided in Supplementary S4 Note).

The occurrence of tears was manipulated between participants, that is, each participant rated either tearful or non-tearful targets. The remaining factors were manipulated within participants. Each participant was presented with five different targets (one neutral, one high-warmth, one low-warmth, one high-competence, and one low-competence) in random order. For each target, situational context and target gender were decided at random. Participants' task was to rate each target on several measures. After rating all targets, participants completed a measure of the Dark Triad and provided demographic information.

**Materials. Pictures of targets:** We used pictures of 30 targets (15 men and 15 women) taken from the Basel Face Database [43]. The database consists of standardized posed portrait photographs of real people that were systematically digitally altered to model specific facial features and, in consequence, create faces appearing more or less extreme on the facial warmth and competence dimensions. For each target, we used the original (neutral) picture and its four modified versions (low-warmth, high-warmth, low-competence, and high-competence). We created a tearful variant of each picture by adding tears digitally (see Fig 3), which resulted in a total of 300 pictures.

**Table 1. Overview of Demographic Information and Exclusions Across the Four Samples.**

| Country | N | Exclusions | | | | Final N | Gender | Age |
|---|---|---|---|---|---|---|---|---|
| | | Speeders | Age < 18 | Attention Check | Comprehension Check | | | |
| South Africa | 502 | 1 | 0 | 13 | 0 (31 obs) | 488 | Women = 242, Men = 241, nb = 2 | $M = 28.7$, $SD = 7.6$, 19–66 |
| Canada | 501 | 1 | 0 | 2 | 0 (4 obs) | 498 | Women = 243, Men = 244, nb = 8, other = 3 | $M = 36.2$, $SD = 12.4$, 18–78 |
| Poland | 505 | 3 | 0 | 2 | 0 (5 obs) | 500 | Women = 237, Men = 246, nb = 13, other = 3 | $M = 26.8$, $SD = 8.05$, 18–68 |
| Norway | 506* | 9 | 0 | 90 | 10 (166 obs) | 407 | Women = 204, Men = 200, nb = 3 | $M = 47.5$, $SD = 15.9$, 19–85 |

*Note.* obs = observations, nb = non-binary. *Recorded 775 responses, but only 506 were complete.

**Situational vignettes:** To manipulate situational context, we used 10 vignettes depicting the targets in various manipulative situations and 10 non-manipulative versions of these situations. In manipulative situational contexts, the target was always presented as someone trying to exert social influence by changing another person's behavior or decision (e.g., "X was trying to make a medical receptionist let them jump a waiting list to see a doctor"; "X was trying to make their partner let them adopt a dog"), while in non-manipulative situational contexts, the situation involved the same person but no attempt to influence this person was made (e.g., "X was waiting to see a doctor and was talking with a medical receptionist"; "X wanted to adopt a dog and was telling their partner about it").

The vignettes were developed using a three-step systematic approach (for details, see Supplementary S3 Note). First, we asked 250 participants (122 men, 125 women, 3 other, $M_{age}$ = 40.58) to recall and describe real-life situations in which they experienced being manipulated with tears/crying or used tears/crying for manipulative purposes themselves. Based on the 305 episodes participants provided, we created an initial pool of 32 vignettes (see Supplementary Table S2). Second, we asked another sample of participants ($n$ = 113; 55 men, 50 women, 4 other, 4 provided no answer; $M_{age}$ = 38.2) on three dimensions: manipulativeness, tears/crying probability, and morality. We added morality ratings because perceived morality and honesty are inherently connected [62], and thus, perceived morality may strongly bias the perception of the target's honesty (hence, our aim was to exclude vignettes describing explicitly morally wrong or good situations). Ratings were made on a 5-point scale from *very unlikely* or *morally wrong* (1) to *very likely* or *morally right* (5). Based on the ratings, we selected 10 vignettes that scored high on manipulativeness and tears/crying probability, and medium on morality. Third, we created a non-manipulative variant of each of the 10 selected vignettes.

**Measures.** For each target, participants completed the exact same measures, rating their intention to support the target, the target's perceived characteristics and emotional expression, and their own felt emotions. All measures were rated on a 7-point scale ranging from *not at all* (1) to *very much* (7). Participants from Canada and South Africa completed all measures in English, while participants from Norway and Poland completed translations of these measures. Most of these translations were already used in the CCT project [3]. The remaining measures were translated into Polish and

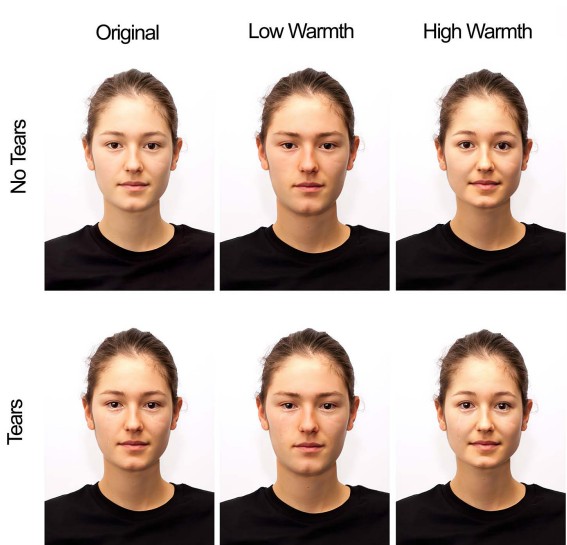

**Fig 3. Overview of Main Manipulation of Face Warmth as based on Walker et al. (2018) and Tears Digitally Added.** Copyright by the Basel Face Database [43].

Norwegian by psychologists speaking both English and Polish/Norwegian (except the Dirty Dozen, in which case we used the Polish adaptation [63]). All discrepancies between the translations were discussed before creating the final measures.

**Support intentions:** To assess support intentions, we used two items ("I would offer support to this person"; "I would be there if this person needed me") from the CCT project [3].

**Perceived honesty, manipulativeness, warmth, and competence:** We assessed the perceived honesty of the target using three items ("honest", "sincere", and "reliable") from a previous study on crocodile tears [4] and the perceived manipulativeness of the target using two items ("manipulative" and "knowing how to behave to get what he/she wants" [64]). For perceived warmth ("warm" and "friendly") and competence ("competent" and "capable"), we applied four items from the CCT project [3].

**Perceived expression authenticity and appropriateness:** The perceived authenticity of the target's expression was assessed with two items ("authentic" and "genuine" [65]). We also asked participants about perceived expression appropriateness ("How appropriate do you think the expression of the person is?" [3]).

**Perceived helplessness and affective reactions:** We also measured additional variables that were not focal to the current research, using items from the CCT project [3]. Participants rated the perceived helplessness ("helpless" and "overwhelmed") of the target as well as their own affective reactions: felt compassion ("compassionate" and "soft-hearted") and distress ("disturbed" and "upset").

**Dark Triad:** To assess participants' Dark Triad personality traits (Machiavellianism, narcissism, and psychopathy), we used the Dirty Dozen [66], a 12-item instrument, using a 5-point response scale from *not like me at all* (1) to *very much like me* (5).

## Results

All analyses were conducted using R (R Project, 2023, version 4.4.1). For main analyses, we employed the following packages: *tidyverse* (version 1.3.2 [67]), *lme4* (version 1.1–32 [68]), *emmeans* (version 1.8.5 [69]), *sjPlot* (version 2.8.14 [70]). Most analyses (if not stated differently) employed multilevel models with random effects according to participants nested in the country and random intercepts for the different pictures. For all analyses, the alpha level was set at .05.

We computed scores for the different variables by averaging item scores. An overview of descriptives and internal reliability statistics (correlation coefficients for two items and Cronbach's *alphas* for more than two items) is provided in Table 2. Internal reliability was acceptable except for the perceived manipulativeness measure. The two items, "manipulative" and "knows how to behave to get what he/she wants", correlated only to a medium degree across the four samples (*r* between .34 and .54). Therefore, against the preregistration, we decided to keep only "manipulative" as a measure of perceived manipulativeness due to its higher face validity. Additional analyses, including both items, are presented in Supplementary S4 Note.

**Confirmatory analyses.** Below, we report the confirmatory analyses as detailed in the preregistration. Deviations from the original preregistration are explicitly noted.

**Manipulation check:** We first checked whether the manipulated variables were perceived as intended. For face warmth and face competence (Supplementary S4 Note), we only focused on faces without tears. This decision was not pre-registered.

**Perceived warmth:** We computed a multilevel model with perceived warmth as the outcome variable and a factor coding for whether the dimension included a low-warmth face (−1), neutral face (0), or high-warmth face (1). An overview is presented in Supplementary Table S4. High-warmth faces (*M* = 4.19, *SE* = .06) were perceived as warmer than low-warmth faces (*M* = 3.12, *SE* = .06, *d* = .68 [.61,.76], *p* < .001) and neutral faces (*M* = 3.81, *SE* = .06, *d* = .24 [.16,.32], *p* < .001). We observed more variation between participants than between countries. We followed up by exploring a potential interaction with occurrence of tears and observed a statistically significant interaction effect (Supplementary Table S5). Adding tears increased perceptions of warmth for low-warmth faces (no-tears: *M* = 3.13, *SE* = 0.06; tears: *M* = 3.58, *SE* = 0.06), had

**Table 2. Overview of reliability statistics, mean values, and standard deviations for each measure per country (Studies 1 and 2).**

| Measure/Country | South Africa n (2396–2409) | | | Canada n (2486) | | | Poland n (2494–2495) | | | Norway n (1907–1909) | | | United Kingdom n (1591–1594) | | |
|---|---|---|---|---|---|---|---|---|---|---|---|---|---|---|---|
| | r/α | M | SD | r/α | M | SD | r/α | M | SD | r/α | M | SD | r/α | M | SD |
| Perceived Appropriateness | – | 4.30 | 1.90 | – | 4.12 | 1.70 | – | 3.77 | 1.73 | – | 4.18 | 1.72 | – | 4.62 | 1.60 |
| Perceived Authenticity | .86 | 4.32 | 1.82 | .88 | 4.18 | 1.58 | .89 | 3.94 | 1.69 | .87 | 4.28 | 1.65 | .88 | 4.63 | 1.60 |
| Perceived Manipulativeness | .34 | 4.26 | 1.52 | .45 | 4.02 | 1.31 | .54 | 3.98 | 1.34 | .49 | 3.93 | 1.46 | – | 3.40 | 1.58 |
| Cognitive Inferences | | | | | | | | | | | | | | | |
| Perceived Competence | .68 | 4.34 | 1.46 | .80 | 4.11 | 1.22 | .75 | 3.66 | 1.21 | .81 | 4.03 | 1.45 | .80 | 4.37 | 1.20 |
| Perceived Warmth | .80 | 3.78 | 1.68 | .85 | 3.60 | 1.42 | .83 | 3.66 | 1.39 | .86 | 3.78 | 1.62 | .81 | 3.99 | 1.38 |
| Perceived Honesty | .80 | 4.14 | 1.69 | .86 | 3.95 | 1.44 | .81 | 3.71 | 1.38 | .86 | 4.07 | 1.65 | .83 | 4.49 | 1.41 |
| Perceived Helplessness | .72 | 3.33 | 1.81 | .69 | 3.16 | 1.53 | .77 | 3.47 | 1.66 | .74 | 3.58 | 1.69 | – | 3.68 | 1.61 |
| Affective Reactions | | | | | | | | | | | | | | | |
| Felt Compassion | .86 | 3.71 | 1.84 | .88 | 3.31 | 1.61 | .75 | 2.68 | 1.51 | .87 | 3.47 | 1.81 | .87 | 4.07 | 1.63 |
| Felt Distress | .62 | 2.7 | 1.65 | .58 | 2.54 | 1.41 | .53 | 2.72 | 1.44 | .79 | 2.91 | 1.72 | .52 | 2.82 | 1.45 |
| Motivational Outcomes | | | | | | | | | | | | | | | |
| Support Intentions | .93 | 4.26 | 1.86 | .93 | 3.82 | 1.56 | .90 | 3.60 | 1.57 | .89 | 3.90 | 1.75 | – | 4.59 | 1.56 |
| Dark Triad | | | | | | | | | | | | | | | |
| Psychopathy | .70 | 2.29 | 1.13 | .78 | 2.51 | 1.17 | .79 | 2.58 | 1.25 | .82 | 2.61 | 1.39 | .80 | 2.41 | 1.16 |
| Narcissism | .85 | 2.86 | 1.46 | .85 | 2.81 | 1.32 | .86 | 3.31 | 1.44 | .89 | 2.91 | 1.53 | .86 | 2.53 | 1.24 |
| Machiavellianism | .82 | 2.64 | 1.35 | .86 | 2.28 | 1.15 | .87 | 2.57 | 1.34 | .89 | 2.44 | 1.44 | .86 | 2.15 | 1.10 |

no effect for neutral faces (no-tears: $M = 3.82$, $SE = 0.06$; tears: $M = 3.82$, $SE = 0.06$), and reduced perceived warmth for high-warmth faces (no tears: $M = 4.19$, $SE = 0.06$; tears: 3.91, $SE = 0.06$). This interaction analysis was not pre-registered.

**Perceived manipulativeness:** Finally, we tested whether targets depicted in manipulative contexts were perceived as more manipulative than targets depicted in non-manipulative contexts. We observed a significant difference for the item "manipulative" ($d = .42$ [.34,.49], $p < 0.001$). Results for the two-item index are presented in Supplementary S4 Note. We explored the interaction between manipulative context and occurrence of tears and observed a statistically significant interaction effect (Supplementary Table S7). Both manipulative contexts and tears increased perceived manipulativeness on average, thereby confirming that the situational context manipulation was successful. However, adding tears resulted in a stronger increase in perceived manipulativeness for manipulative contexts (no-tears: $M = 3.99$, $SE = 0.07$; tears: $M = 4.44$, $SE = 0.07$) compared to non-manipulative contexts (no-tears: $M = 3.40$, $SE = 0.07$; tears: $M = 3.58$, $SE = 0.07$).

**H1. Perceptions of honesty:** We ran a multilevel model with perceived honesty as the outcome variable and factors coding for occurrence of tears ($-.5$ = no tears, $.5$ = tears), face warmth ($-.5$ = low warmth, $.5$ = high warmth), and situational context ($-.5$ = non-manipulative, $.5$ = manipulative). An overview is presented in Table 3 and Fig 4.

We observed a significant main effect, suggesting that perceived honesty increased for tearful targets ($d = .17$ [.05,.30], $p = .007$). As expected, this effect was moderated by whether the face was low or high in warmth, but the direction of this effect was contrary to our hypothesis: Tears increased perceived honesty for faces low in warmth ($d = .17$ [.06,.27]), but decreased it for faces high in warmth ($d = -.16$ [$-.26$, $-.06$], Fig 4, H1).

We also observed that faces high in warmth were rated as more honest than faces low in warmth ($d = .19$ [.07,.31], $p = .001$), and targets pictured in manipulative situational contexts were perceived as less honest than targets pictured in non-manipulative situational contexts ($d = -.40$ [$-.52$, $-.28$], $p < .001$). Additionally, we observed a significant interaction between face warmth and situational context. Presenting the targets in manipulative situations decreased honesty more strongly for faces low in warmth ($d = -.51$ [$-.61$, $-.41$]) than faces high in warmth ($d = -.36$ [$-.46$, $-.26$]). We observed no statistically significant interaction between tears and situational context or a statistically significant three-way interaction.

**Table 3. Overview of the model with occurrence of tears, face warmth, and situational context on perceived honesty, perceived target manipulativeness, and perceived expression authenticity.**

| Predictors | Perceived Honesty | | | | | Perceived Target Manipulativeness | | | | | Perceived Expression Authenticity | | | | |
|---|---|---|---|---|---|---|---|---|---|---|---|---|---|---|---|
| | Estimates | Std. Beta | CI | Std. CI | p | Estimates | Std. Beta | CI | Std. CI | p | Estimates | std. Beta | CI | Std. CI | p |
| (Intercept) | 4.02 | 0.07 | [3.81; 4.24] | [−0.07, 0.20] | **<.001** | 3.58 | −0.18 | [3.40, 3.76] | [−0.28, −0.07] | **< 0.001** | 4.37 | 0.13 | [4.15, 4.59] | [−0.00, 0.26] | **<.001** |
| Occurrence of Tears | 0.27 | 0.17 | [0.07, 0.46] | [0.05, 0.30] | **.007** | 0.07 | 0.04 | [−0.12, 0.26] | [−0.07, 0.15] | 0.475 | 0.17 | 0.10 | [−0.01, 0.36] | [−0.01, 0.21] | .069 |
| Face Warmth | 0.30 | 0.19 | [0.11, 0.48] | [0.07, 0.31] | **.001** | −0.32 | −0.19 | [−0.46, −0.18] | [−0.27, −0.11] | **< 0.001** | 0.02 | 0.01 | [−0.11, 0.16] | [−0.07, 0.09] | .745 |
| Situational Context | −0.62 | −0.40 | [−0.80, −0.43] | [−0.52, −0.28] | **<.001** | 0.63 | 0.36 | [0.48, 0.78] | [0.27, 0.45] | **< 0.001** | −0.34 | −0.20 | [−0.49, −0.20] | [−0.29, −0.12] | **<.001** |
| Occurrence of Tears×Face Warmth | −0.33 | −0.21 | [−0.59, −0.07] | [−0.38, −0.05] | **.012** | 0.23 | 0.13 | [0.03, 0.43] | [0.02, 0.25] | **0.022** | −0.25 | −0.15 | [−0.43, −0.06] | [−0.26, −0.03] | **.011** |
| Occurrence of Tears × Situational Context | −0.10 | −0.06 | [−0.36, 0.16] | [−0.24, 0.11] | .459 | 0.19 | 0.11 | [−0.03, 0.40] | [−0.02, 0.23] | 0.089 | −0.29 | −0.17 | [−0.50, −0.09] | [−0.30, −0.05] | **.006** |
| Face Warmth × Situational Context | 0.29 | 0.19 | [0.03, 0.55] | [0.02, 0.36] | **.031** | | | | | | | | | | |
| Occurrence of Tears × Face Warmth × Situational Context | −0.19 | −0.12 | [−0.56, 0.19] | [−0.36, 0.12] | .324 | | | | | | | | | | |
| **Random Effects** | | | | | | | | | | | | | | | |
| $\sigma^2$ | 1.70 | | | | | 2.36 | | | | | 2.13 | | | | |
| $\tau_{00}$ | 0.52 ID:Country | | | | | 0.49 ID:Country | | | | | 0.60 ID:Country | | | | |
| | 0.02 pic_id | | | | | 0.02 pic_id | | | | | 0.05 pic_id | | | | |
| | 0.02 Country | | | | | 0.01 Country | | | | | 0.03 Country | | | | |
| ICC | 0.25 | | | | | 0.18 | | | | | 0.24 | | | | |
| N | 1883 ID | | | | | 1883 ID | | | | | 1883 ID | | | | |
| | 4 Country | | | | | 4 Country | | | | | 4 Country | | | | |
| | 30 pic_id | | | | | 30 pic_id | | | | | 30 pic_id | | | | |
| Observations | 3720 | | | | | 3723 | | | | | 3724 | | | | |
| Marginal R²/ Conditional R² | 0.047/ 0.284 | | | | | 0.055/ 0.226 | | | | | 0.026/ 0.259 | | | | |

*Note.* Occurrence of tears (−.5 = no tears, .5 = tears); face warmth (−.5 = low warmth, .5 = high warmth); situational context (−.5 = non-manipulative, .5 = manipulative). Std. = standardized.

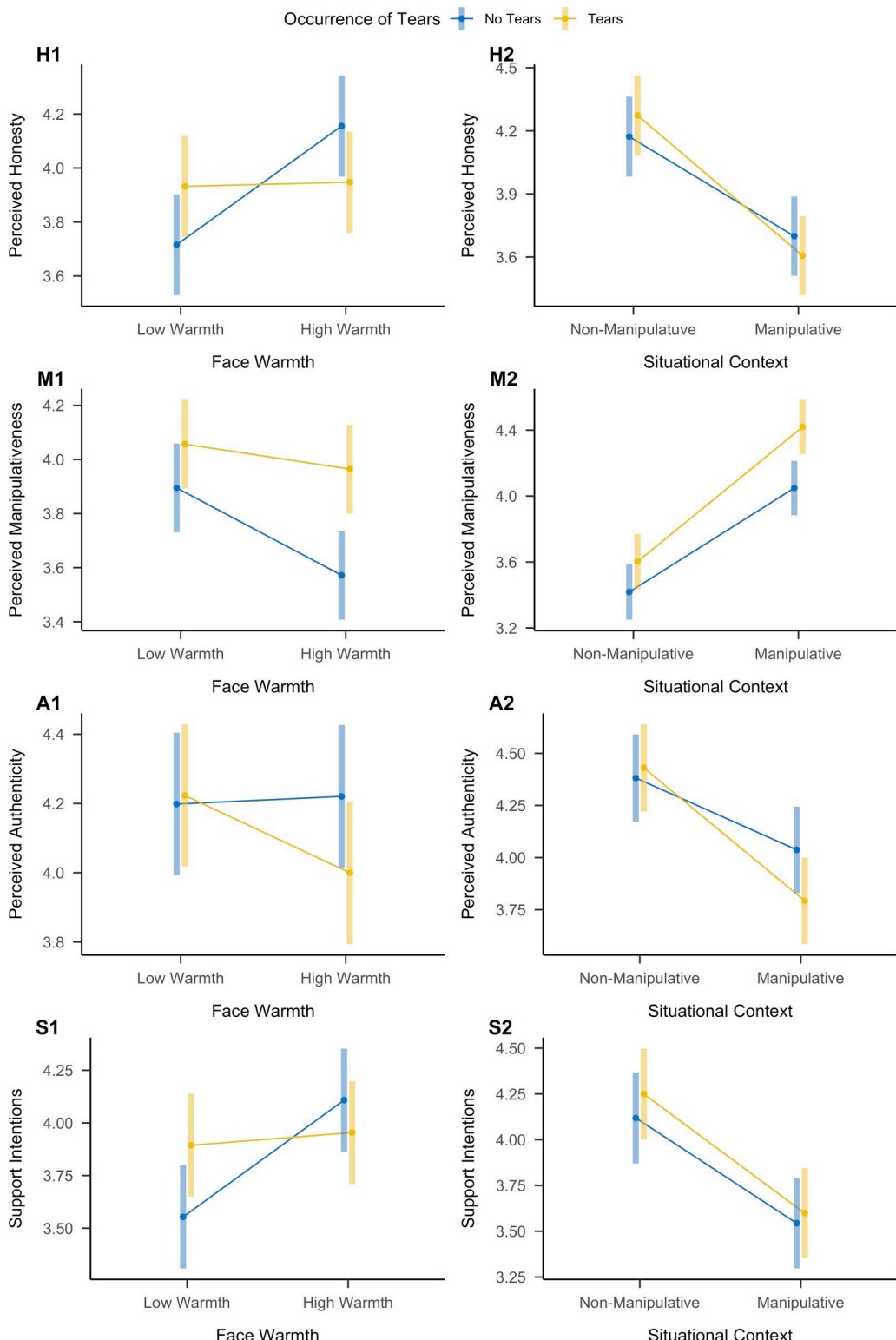

**Fig 4. Overview of Interaction Effects Between Occurrence of Tears and Face Warmth (First Column) and Occurrence of Tears and Situational Context (Second Column) on Perceived Honesty (First Row, H), Perceived Manipulativeness (Second Row, M), Perceived Authenticity (Third Row, A), and Support Intentions (Fourth Row, S). Values present predicted marginal effects, and error bars represent 95% confidence intervals.**

As expected, tears reduced honesty more strongly for manipulative situations ($d = -.07$ [$-.18, .04$]) in comparison to non-manipulative situations ($d = .08$ [$-.03, .19$], Fig 4, H2), but this effect was small and not statistically significant.

**H2. Moderated mediation by perceived expression authenticity and perceived target manipulativeness:** For H2, we tested whether the effect of occurrence of tears on perceived honesty was mediated by the perceived manipulativeness of the target and perceived authenticity of the expression, and whether these effects were moderated by face warmth and situational context. In total, we ran three multilevel models. In the first model, we regressed tears, and the interaction of tears with face warmth and situational context on perceived manipulativeness (the single item). In the second model, we regressed the same factors on perceived expression authenticity. An overview is presented in Table 3.

For perceived target manipulativeness, we observed no statistically significant main effect of tears. Specifically, tears increased perceptions of target manipulativeness ($d = .04$ [$-.07, .15$]), but this effect was small and not statistically significant. However, we observed a significant interaction between tears and face warmth. Tears increased perceived target manipulativeness, but this effect was stronger for high-warmth faces ($d = .26$ [$.16, .36$]) than low-warmth faces ($d = .11$ [$.01, .21$], Fig 4, M1). Although the interaction between tears and situational context was not statistically significant, we observed that tears increased perceived target manipulativeness slightly stronger for manipulative situations ($d = .24$ [$.14, .34$]) than for non-manipulative ones ($d = .12$ [$.01, .23$], Fig 4, M2).

For perceived expression authenticity, we observed again no statistically significant main effect of tears: Tears slightly increased perceived authenticity ($d = .10$ [$-.01, .21$]), but the effect was small. However, we observed significant interaction effects of tears and situational context, as well as tears and face warmth. Tears decreased perceived expression authenticity for faces high in warmth ($d = -.15$ [$-.25, .05$]), but this was not the case for faces low in warmth ($d = .02$ [$-.09, .12$], Fig 2, A1). Moreover, tears decreased perceived expression authenticity for manipulative situations ($d = -.17$ [$-.27, -.06$]), but not for non-manipulative situations ($d = .03$ [$-.08, .14$], Fig 4, A2).

In the third model, we regressed tears, perceived authenticity (centered), and perceived manipulativeness (centered) on perceived honesty. An overview is provided in Supplementary Table S8. We observed a significant effect of tears on perceived honesty, with tears slightly increasing perceived honesty ($d = .08$ [$.03, .12$]). In addition, we observed that perceived expression authenticity positively predicted perceived honesty ($\beta = .66$ [$.64, .68$]), while perceived target manipulativeness predicted it negatively ($\beta = -.21$ [$-.23, -.19$]).

Next, we tested for indirect effects of tears on perceived honesty via perceived target manipulativeness and perceived expression authenticity as moderated by face warmth and situational context. We calculated confidence intervals using a Monte Carlo estimation [71]. An overview is presented in Table 4. We observed significant indirect effects of tears on perceived honesty via perceived manipulativeness for all levels. Faces high in warmth and manipulative situations showed slightly stronger indirect effects than faces low in warmth and non-manipulative situations. Tears increased perceived manipulativeness somewhat more for high-warmth faces and manipulative situations. In turn, perceived manipulativeness was negatively correlated with perceived honesty. Perceived authenticity showed significant indirect effects only for faces high in warmth and manipulative situations. Tears reduced perceived expression authenticity for faces high in warmth and manipulative situations. In turn, perceived expression authenticity was positively correlated with perceived honesty.

**H3. Mediation by perceived honesty on social support intentions:** Next, we investigated whether perceived honesty mediated the effect of tears on social support intentions. We observed no statistically significant direct effect of tears on social support intentions ($d = .05$ [$-.02, .13$], $p = .150$). When exploring this relationship, we observed a significant moderation by face warmth. Tears increased support intentions for low-warmth faces ($d = .25$ [$.13, .36$]) but slightly decreased these intentions for high-warmth faces ($d = -.11$ [$-.22, -.001$], Fig 4S1; Supplementary Table S9).

We observed a significant indirect effect of tears on social support intentions via perceived honesty, $B = .21$ [$.06, .37$]. Tears increased perceived honesty (when controlling for face warmth and situational context), which was

**Table 4. Unstandardized (Studies 1 & 2) and standardized (meta-analysis) indirect effects for perceived target manipulativeness and perceived expression authenticity on perceived honesty and for perceived honesty on support intentions when controlling for the moderators.**

| Outcome | Mediator | Moderator | Moderator Level | Study 1 | | Study 2 | | Meta Analysis | |
|---|---|---|---|---|---|---|---|---|---|
| | | | | **B** | 95% CI | **B** | 95% CI | β | 95% CI |
| Perceived Honesty | Perceived Target Manipulativeness | Face Warmth/ Target Gender | Low Warmth/Male | −.03 | [−.06, −.002] | .004 | [−.06,.06] | −.02 | [−.03, −.01] |
| | | | High Warmth/Female | −.07 | [−.10, −.05] | −.04 | [−.10,.02] | −.02 | [−.04, .01] |
| | | Situational Context | Non-Manipulative | −.03 | [−.06, −.004] | −.01 | [−.06,.04] | −.01 | [−.01,.00] |
| | | | Manipulative | −.07 | [−.10, −.04] | −.04 | [−.10,.03] | −.03 | [−.05, −.01] |
| | | None | | −.01 | [−.05,.02] | −.02 | [−.06,.02] | −.02 | [−.03, −.01] |
| | Perceived Expression Authenticity | Face Warmth/ Target Gender | Low Warmth/Male | .01 | [−.07,.11] | .08 | [−.04,.20] | .00 | [−.03,.04] |
| | | | High Warmth/Female | −.13 | [−.22, −.04] | .03 | [−.08,.15] | −.05 | [−.08,.02] |
| | | Situational Context | Non-Manipulative | .03 | [−.07,.13] | .02 | [−.09,.13] | −.01 | [−.03,.01] |
| | | | Manipulative | −.15 | [−.24, −.06] | .07 | [−.04,.19] | −.04 | [−.08,.01] |
| | | None | | .10 | [−.008,.22] | .05 | [−.03,.13] | −.01 | [−.04,.02] |
| Support Intentions | Perceived Honesty | Face Warmth/ Target Gender | Low Warmth/Male | .16 | [.06,.28] | .17 | [.02,.32] | .05 | [.02,.08] |
| | | | High Warmth/Female | −.15 | [−.27, −.06] | .09 | [−.06,.23] | −.05 | [−.09, −.02] |
| | | Situational Context | Non-Manipulative | .08 | [−.03,.20] | .06 | [−.07,.18] | .00 | [−.02,.02] |
| | | | Manipulative | −.07 | [−.18,.04] | .17 | [.01,.33] | −.02 | [−.06,.02] |
| | | None | | .21 | [.06,.37] | .12 | [.01,.23] | .01 | [−.01,.03] |

associated with social support intentions (Table 4). The indirect effect was positive for faces low in warmth, $B = .16$ [.06,.28], but negative for faces high in warmth, $B = −.15$ [−.27, −.06]. Faces low in warmth with tears increased perceptions of honesty, which was associated with support intentions, while faces high in warmth reduced perceptions of honesty. We also explored the indirect effect of situational context. Non-manipulative vignettes showed an indirect effect, $B = .08$ [−.03, .20], while manipulative vignettes showed a direct effect, $B = −.07$ [−.18, .04], though not statistically significant (Table 4).

**H4. Moderation by the Dark Triad:** We repeated a multilevel model with perceived honesty as the outcome variable, tears and situational context as the predictors, as well as narcissism, psychopathy, and Machiavellianism (all centered) and their interactions. An overview is provided in Supplementary Table S10. We observed no significant main or interaction effects of any of the Dark Triad personality traits, nor the predicted three-way interaction with tears, situational contexts, and the Dark Triad.

**Exploratory analyses. Moderation by the Dark Triad:** As we did not find any significant effects when testing H4, we further explored the impact of the dark personality traits by running a multilevel model with perceived honesty as the outcome variable, tears, and situational contexts as predictors, and each trait (centered) separately (Supplementary Table S11). We observed significant interactions between tears and psychopathy (β = −.08, $p = .012$) and between tears and Machiavellianism (β = −.07, $p = .048$). Tears reduced perceived honesty for increasing levels of psychopathy, and the same was true for Machiavellianism (Fig 5). Further, we explored the impact of personality traits on target perceived manipulativeness (Supplementary Table S12). Participants rated targets as more manipulative when these targets were depicted in manipulative than non-manipulative situational contexts, but this difference became smaller with increasing levels of narcissism and Machiavellianism (Supplementary Fig S4).

**Impact of target gender:** We also explored the impact of target gender on perceptions of honesty (Supplementary S4 Note). We observed a statistically significant two-way interaction between occurrence of tears and target gender. Tears slightly increased perceptions of honesty for male targets ($d = .10$ [.01,.21]), but they slightly decreased these perceptions for female targets ($d = −.10$ [−.20,.01], Fig S5, Supplementary Table S13). In addition, we observed a statistically significant two-way interaction between situational context and target gender. Manipulative situational contexts reduced

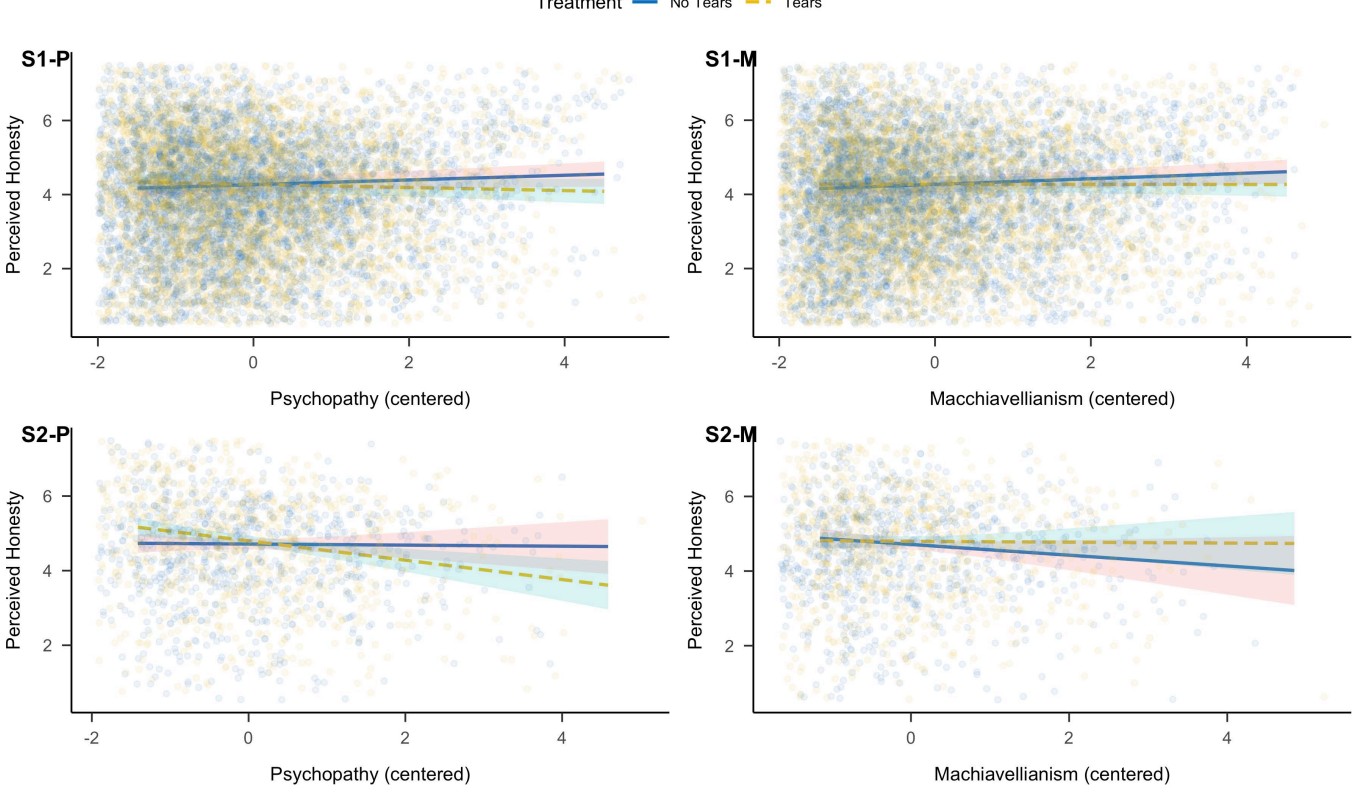

**Fig 5. Overview of Interactions between Psychopathy (P), Machiavellianism (M), and Occurrence of Tears on Perceived Honesty for Study 1 (S1, first row) and Study 2 (S2, second row).** Confidence bands represent 95% confidence intervals.

perceptions of honesty stronger for male targets ($d=-.52$ [−.62, −.42]) compared to female targets ($d=-.36$ [.26, −.46]), which was not observed in non-manipulative contexts (Supplementary Table S13).

Similarly, we explored the impact of target gender on ratings of target manipulativeness and expression authenticity. For target manipulativeness, we observed a significant main effect of target gender, with male targets being perceived as more manipulative than female targets ($d=-.15$ [−.25, −.05]) and significant interaction with occurrence of tears, suggesting that tears increased perceived manipulativeness more strongly for female targets ($d=.25$ [.15,.36]) than for male targets ($d=.11$ [.01,.21]). This was driven by the fact that male targets were perceived as more manipulative when they were presented without tears than with tears (Fig S5, Supplementary Table S14).

Finally, we tested the impact of target gender on the relationship between occurrence of tears and support intentions (Supplementary Table S16). We observed both a statistically significant main effect of tears and an interaction effect between tears and target gender. Tears, in contrast to no tears, increased support intentions for male targets ($d=.18$ [.07,.30]), but not for female targets ($d=-.05$ [−.17,.07], Fig S5). When testing the mediation by perceived honesty separately for male and female targets, we observed that perceived honesty showed a stronger mediation effect for male ($B=.32$ [.09,.56]) compared to female targets ($B=.12$ [−.09,.34]).

**Impact of perceived appropriateness:** We also explored the role of perceived appropriateness (Supplementary S4 Note). We found that expressions of the targets depicted in non-manipulative contexts were perceived as more appropriate than expressions of the targets depicted in manipulative ones ($d=-.41$ [−.54, −.28]). We also observed a statistically significant three-way interaction. Shedding tears increased perceptions of appropriateness for low-warmth targets in manipulative contexts, but not for high-warmth targets or non-manipulative contexts. (Supplementary Table S17, Fig S5).

In addition, we explored the moderation effect of perceived expression appropriateness on honesty, manipulativeness, authenticity, and support intentions. For all measures, we observed a statistically significant main effect of perceived appropriateness. Higher appropriateness was associated with higher perceived honesty, authenticity, and support intentions, and lower perceived manipulativeness. We also observed that this effect was not moderated by face warmth or manipulative context (Supplementary Table S18, Fig S6).

**Impact of country:**   While we did not observe much variation due to the different countries in the main models, we explored the impact of the country. We created a factor coding for countries with *higher* (Canada and Norway) and *lower trust levels* (South Africa and Poland; based on different indices; see Supplementary S2 Note). We then ran four multilevel models with perceived honesty, manipulativeness, expression authenticity, and support intentions as the DVs and occurrence of tears, the country index, and their interaction. We only observed a statistically significant interaction for perceived manipulativeness (Supplementary Table S20). For low-trust countries, tears increased perceptions of manipulativeness more strongly ($d = .29$ [.21,.36]) than for high-trust countries ($d = .12$ [.04,.21]). This effect was driven by the fact that high-trust countries perceived tearful targets as less manipulative than low-trust countries (Fig 6).

**Impact of presentation order:**   Finally, driven by the observation that rating five targets in a row might have caused fatigue, we explored possible order effects by testing ratings of perceived honesty, perceived manipulativeness, perceived authenticity, and support intentions in a model including occurrence of tears, face warmth, situational context, presentation order, and its interaction with the other predictors (Supplementary Table S21). We observed statistically significant interactions with presentation order and occurrence of tears for perceived honesty, perceived authenticity, and support intentions. For all three measures, ratings decreased for tearful pictures with increasing presentation order (Supplementary Fig S7). While tearful pictures compared to the non-tearful ones received higher honesty, authenticity, and support intentions ratings for the first stimuli, these ratings decreased for each repetition, with the difference between tearful and non-tearful stimuli being reduced for the last round.

## Discussion

Study 1 confirmed that the perception of tearful targets' honesty is context-dependent, but our findings did not support many of our predictions. Most importantly, against H1, we found that the effects of tears on perceived honesty, manipulativeness, and expression authenticity were more detrimental for faces high than low in warmth. These results challenge the idea that tears shed by warm targets are particularly believable and suggest the opposite. Possibly, when low-warmth targets engage in a warmth-signaling behavior that is considered atypical of them (i.e., tearing up), observers assume that there must be a genuine reason to do so. This might increase honesty ratings, but future research is needed to corroborate these unexpected results. Importantly, these results were driven by the fact that low-warmth targets received lower honesty ratings when shown without tears, and adding tears made them more similar to high-warmth targets.

We also found that targets in manipulative situations were perceived as more manipulative and less honest than targets in non-manipulative situations. However, the interactive effects of situational context and tears on perceived manipulativeness and honesty were non-significant. Interestingly, though, in manipulative situations, in contrast to non-manipulative ones, expressions of tearful (vs. non-tearful) targets were evaluated as less authentic. Overall, these results indicate that tears were perceived as less authentic when shed in manipulative situations, but the role of situational context in moderating target-related inferences (i.e., manipulativeness and honesty) was limited.

Perceived target manipulativeness and expression authenticity were found to mediate the effect of tears on perceived honesty, and this effect was most pronounced when targets were high in facial warmth and the situation was manipulative. Moreover, tears – both directly and indirectly (through perceived honesty) – increased observers' intentions to support low-warmth targets, at the same time decreasing these intentions for high-warmth targets, providing further support for the idea that tears may be particularly beneficial for those less expected to shed them. Notably, this idea aligns with the results of our exploratory analyses, which revealed that tears increased perceived honesty for male targets but decreased

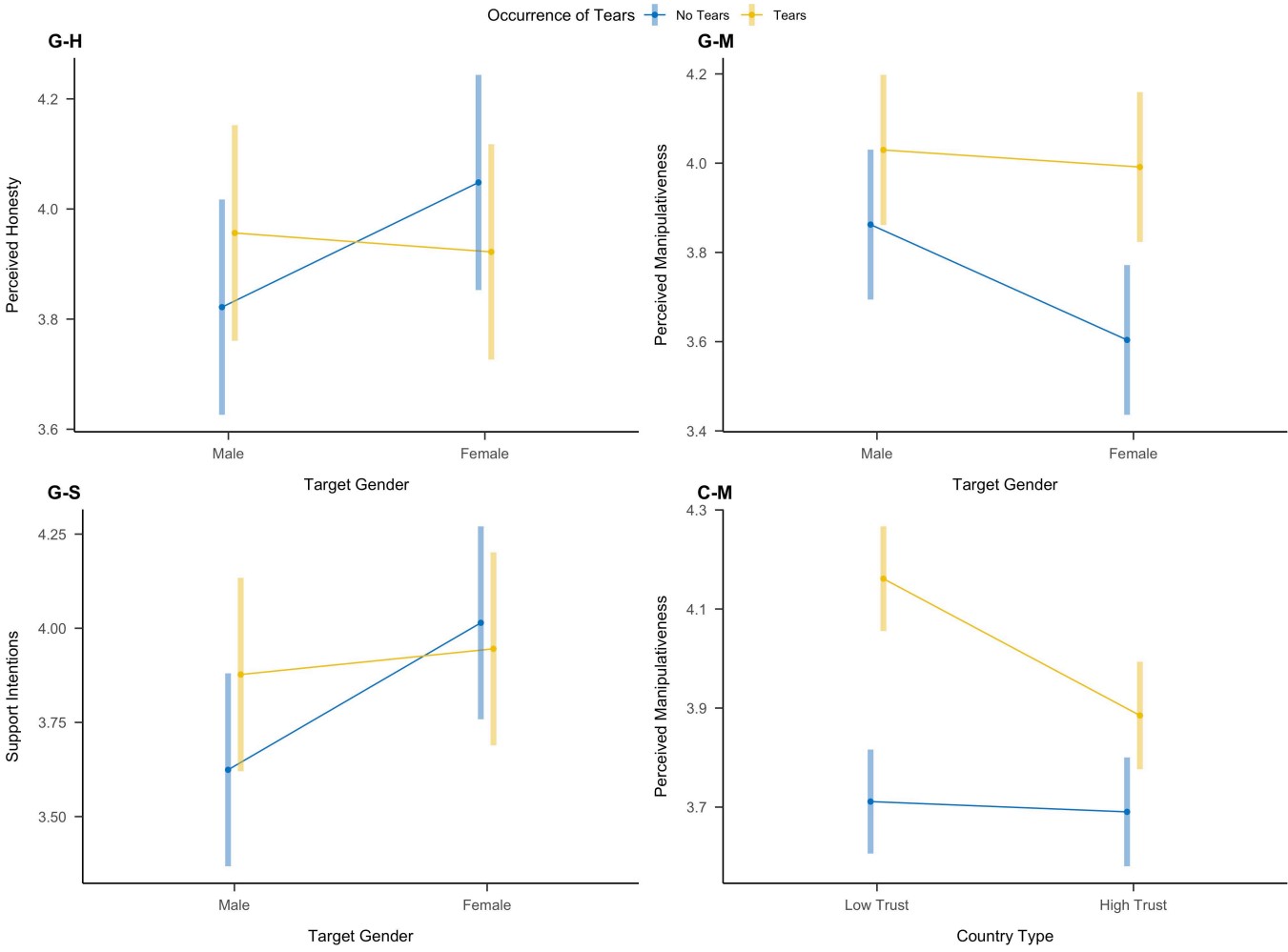

**Fig 6. Overview of Interactions between Occurrence of Tears and Target Gender on Perceived Honesty (G-H), Perceived Manipulativeness (G-M), and Support Intentions (G-S), as well as the Interaction between Country Type and Occurrence of Tears on Perceived Manipulativeness (C-M).** Error bars represent 95% confidence intervals.

it for female targets, and these effects translated into support intentions (tears increased support intentions more strongly for males than for females). These findings converge with the idea that men's tears, similar to low-warmth targets' tears, being less common and less expected, tend to be more socially beneficial [72].

Study 1 showed no support for the hypothesized impact of the observers' Dark Triad traits on honesty ratings, nor the predicted three-way interaction of tears, situational contexts, and the Dark Triad. However, when we explored this relationship for each of the three traits separately, we found that participants high in psychopathy and Machiavellianism evaluated tearful (vs. non-tearful) targets as less honest, which aligns with the notion that the dark aspects of personality reduce the tendency to appreciate the authenticity of others' distress [22].

Further, our exploratory analyses revealed that when the context was manipulative, expressions of high-warmth targets were perceived as more appropriate when they were not showing tears than when they did, while for low-warmth targets, this effect was reversed. Interestingly, we observed no similar pattern in non-manipulative contexts. Moreover, when expressions were perceived as less appropriate (which was particularly the case in manipulative contexts), tears led to

more detrimental effects than not shedding tears. Yet, when expressions were considered appropriate, tears led to more beneficial effects, even when the context was manipulative. This result corroborates accumulating evidence showing that the higher appropriateness is attributed to tears, the more favorable their social effects are (for a review, see [14]).

Finally, the exploratory analyses indicated that shedding tears was associated with perceived manipulativeness more strongly for low-trust countries than high-trust countries, thereby corroborating previous findings indicating that the interpretation of the social signal value of tears might differ cross-culturally [3] and indicating that culture-level trust might be one of the factors contributing to these cultural differences. We did not, however, find similar country effects on perceived honesty, expression authenticity, and support intentions. Hence, the influence of cultural factors on the perception of the tearful targets was limited.

Summing up, Study 1 supported the role of contextual factors in shaping the perceptions of targets' honesty, but many effects we observed were at odds with our hypotheses. Moreover, these effects were small, and the results were complex due to the study design and many moderating factors. We also addressed some relationships only from an exploratory angle and observed a significant reduction in the effects of tears due to exposure to five targets in a row. Given these limitations, we decided to replicate our findings in an independent sample, using more subtle and ecologically valid warmth manipulation and reducing the number of stimuli participants were exposed to.

## Study 2: Replication with different stimuli and sample

Study 1 provided a first test of the circumstances that influence perceptions of honesty for tearful and non-tearful individuals. We obtained evidence that the effect of tears is stronger for low-warmth and male targets. However, the study tested many exploratory analyses focusing on target gender or perceived appropriateness and revealed fatigue effects. Therefore, we conducted a follow-up study to address potential concerns and expand the focus by a) using a more subtle manipulation of perceived warmth based on the target's gender, b) collecting only one rating per participant to avoid order or fatigue effects, c) using a more ecologically valid database of non-standardized pictures, d) registering analyses about the effects of perceived appropriateness, and e) testing the generalizability of results by focusing on another population. For feasibility reasons, we focused on UK-based adults because this sample is well-represented on Prolific and we could easily apply English-language measures we had already used in the previous study. Based on Study 1, we adapted our hypotheses (H1) predicting a stronger tear effect for targets low in warmth (i.e., male targets) and added another hypothesis concerning perceived appropriateness:

H5: We will observe a three-way interaction of the occurrence of tears, manipulative context, and target gender on perceived appropriateness. Appropriateness will be higher for non-manipulative vignettes. For manipulative vignettes, tears will increase appropriateness for male targets but reduce appropriateness for female targets.

### Method

**Participants.** We recruited 1628 UK-based participants via Prolific.com and paid them £0.5 for participation. Based on the preregistered criteria, we excluded participants who a) completed the study faster than ⅓ of the median completion time ($n = 7$), failed an attention check ($n = 7$), failed a comprehension check ($n = 7$), or provided missing data ($n = 12$). The final sample included 1595 participants (788 females, 784 males, 12 non-binary, 6 other, 5 missing) ranging from 18 to 79 years of age ($M = 42.7$, $SD = 13.3$).

We performed a power analysis based on the interaction effect between occurrence of tears and face warmth (H1) in Study 1 using *superpower* [73]. Setting alpha at .05 and power at 80%, we observed a sample size of 419 per cell for the interaction effect. For the total sample, this would mean 1,676 participants. In order to save potential resources, we employed a sequential analysis [74]. In the first step, we recruited 400 participants and tested whether the tear main effect or interaction effect predicted in H1 had reached statistical significance. If this was not the case,

we recruited another 400 participants and checked the results again, and so on, until we reached 1,600 participants. We controlled for the Type-1 error rate by adjusting the alpha level at each step based on a spending function using the GroupSeq package in R (see https://osf.io/qtufw/). Selecting four interim times, two-sided bounds, the alpha level at.05, and a power family function, the spending function suggested an alpha level of .0125, .0161, .0203, and .0248 at the four times. As we did not obtain significant results for the main or interaction effect, we continued to collect $n = 1,600$ participants.

**Procedure.** We employed a 2 occurrence of tears (tears vs. no tears) x 2 target gender (female vs. male) x 2 situational context (non-manipulative vs. manipulative situations) between participants design. In contrast to Study 1, each participant completed ratings for only one target. For each participant, occurrence of tears (no tears: $n = 779$, tears: $n = 816$), situational context (non-manipulative: $n = 765$, manipulative: $n = 830$), and target gender (female: $n = 831$, male: $n = 764$) were decided at random. The study followed the same procedure and measures as in Study 1, except where noted differently.

**Materials. Pictures of targets:** In contrast to Study 1, we used pictures of 10 targets (5 men and 5 women) taken from the Thaulow Refugee Database [15]. The database consists of non-standardized portrait photographs of 26 refugees varying in age and facial appearance. All the portrayed individuals sit in natural positions and show subtle emotional expressions. We selected female targets rated highest on warmth and male targets rated lowest on warmth based on a previous validation study [15]. For each target, we used the original picture and its modified variant with tears added digitally as in previous research [15].

**Situational vignettes:** We employed the same vignettes as in Study 1. We removed one situation ("professor situation", see Supplementary S3 Note) as it was unlikely that some targets would be students, given the broad age range of targets.

**Measures.** We used the exact measures as in Study 1, with three exceptions. For support intentions ("I would offer support to this person"), manipulativeness ("manipulative"), and helplessness ("helpless"), we used only one item to minimize possible fatigue. Reliabilities are presented in Table 2 and S5 Note (Table S22).

## Results

**Confirmatory Results.** For all confirmatory results, we set our alpha level at .025 (see explanation above).

**Manipulation check:** First, we tested whether female targets were perceived as warmer compared to male targets. Performing a Welch's t-test, we observed a statistically significant difference, $t(1586) = 6.12$, $p < .001$, $d = .31$ [.21, .41], with female targets ($M = 4.19$, $SD = 1.38$) rated higher in warmth compared to male targets ($M = 3.77$, $SD = 1.35$). In addition, we observed a statistically significant difference in perceived manipulativeness ratings for vignettes describing manipulative ($M = 3.80$, $SD = 1.59$) and non-manipulative situational contexts ($M = 2.97$, $SD = 1.46$), $t(1593) = 10.9$, $p < .001$, $d = .55$ [.44, .65], confirming the effectiveness of our manipulation.

**H1. Perceptions of honesty:** We ran a linear regression model with perceived honesty as the outcome variable and factors coding for occurrence of tears ($-.5 =$ no tears, $.5 =$ tears), target gender ($-.5 =$ male, $.5 =$ female), and situational context ($-.5 =$ non-manipulative, $.5 =$ manipulative). An overview is presented in Fig 7 and Supplementary Table S23.

The results revealed that perceived honesty increased for tearful targets, but this main effect was non-significant ($d = .09$ [−.12,.29], $p = .397$). We only observed a significant main effect that targets pictured in manipulative situational contexts were perceived on average as less honest than targets pictured in non-manipulative situational contexts ($d = −.50$ [−.70, −.29], $p < .001$). In contrast to Study 1, we observed no statistically significant interactions (Supplementary Table S23, Fig 7). The strongest interaction effect was observed for target gender and situational context, with females being more strongly rated as honest in manipulative situations compared to male targets ($d = .26$ [.12, .40]) in contrast to non-manipulative situations ($d = .08$ [−.06, .21]). Exploring target gender, we also observed that the effect of tears on

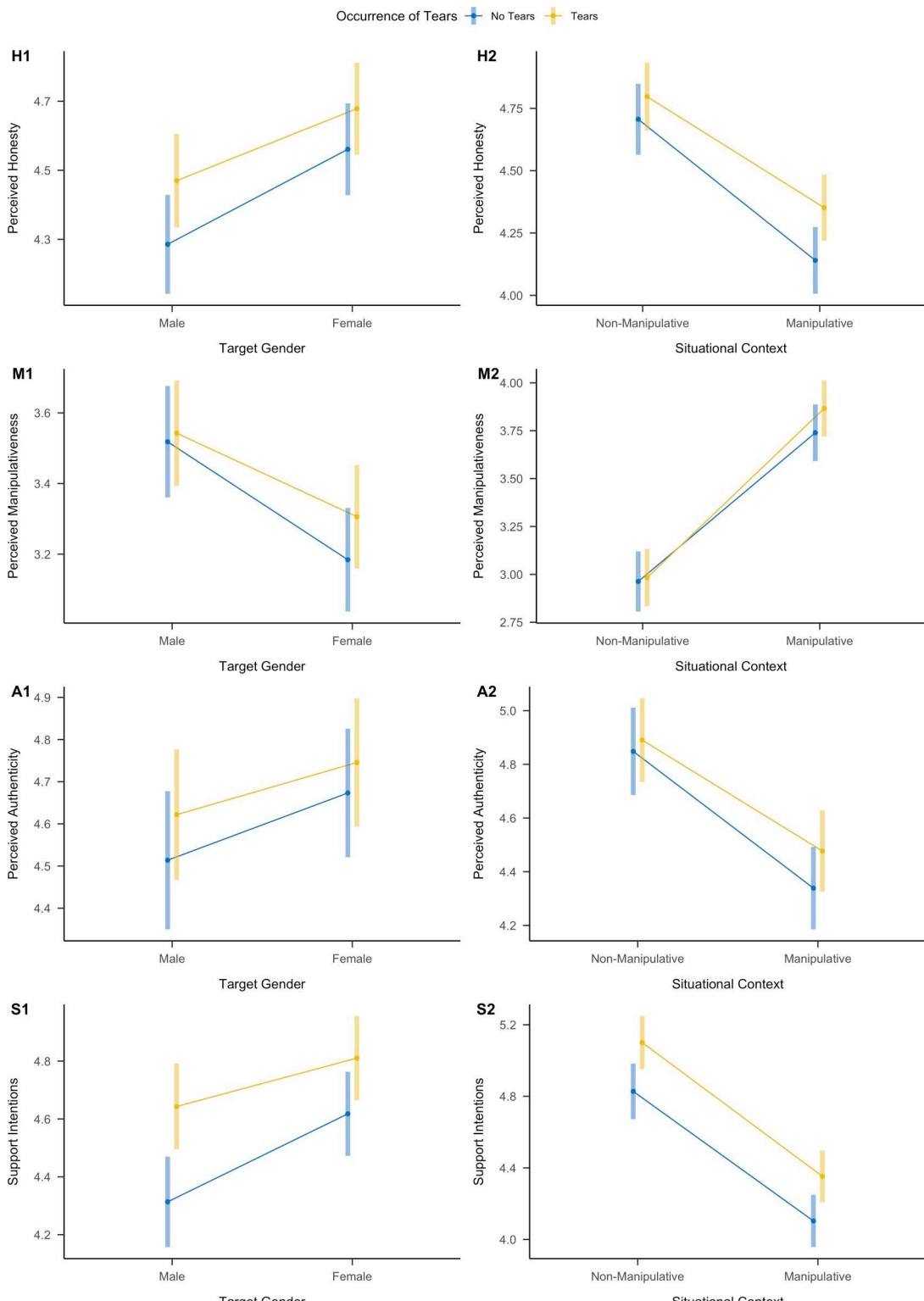

**Fig 7. Overview of Interactions between Occurrence of Tears and Target Gender (First Column) and Occurrence of Tears and Situational Context (Second Column) on Perceived Honesty (H, First Row), Perceived Manipulativeness (M, Second Row), Perceived Authenticity (A, Third Row), and Support Intentions (S, Fourth Row).** Whiskers depict 95% confidence intervals. Effects are predicted marginal effects.

perceived honesty was slightly stronger for male targets ($d = .16$ [.02, .31]) compared to female targets ($d = .08$ [−.05, .22]), but the difference was not statistically significant (Fig 7).

Since none of the interaction effects was statistically significant, we explored the model with only main effects and observed significant effects for occurrence of tears ($d = .11$ [.01, .21], $p = .024$), target gender ($d = .18$ [.08, .27], $p < .001$), and situational context ($d = −.35$ [−.45, −.26], $p < .001$; Supplementary Table S24).

**H2. Mediation via perceived manipulativeness and authenticity:**   We set up a mediation model in *lavaan* with occurrence of tears as the predictor, perceived manipulativeness and authenticity as mediators, and perceived honesty as the outcome. This model was fitted separately for the different levels of target gender and situational context, and bootstrap confidence intervals were computed for the indirect effects using 10,000 repetitions. An overview is presented in Table 4 and Supplementary Table S25. We did not observe a significant indirect effect for any of the combinations. While manipulativeness and authenticity correlated with perceived honesty, occurrence of tears did not influence these ratings (Fig 7; Supplementary Table S25, for detailed results).

**H3. Mediation by perceived honesty on social support intentions:**   We found that tears increased overall social support intentions ($t(1593) = 3.43$, $d = .17$ [.07, .27], $p < .001$), and this effect was slightly stronger for male targets ($d = .24$ [.10, .40]) compared to female targets ($d = .12$ [−.02, .26]; Fig 7). Further exploring this effect suggested that it was strongest for male targets in manipulative contexts ($d = .28$ [.07, .47]) and weakest for female targets in manipulative contexts ($d = .04$ [−.14, .23]; Supplementary Table S26; Supplementary Fig S8).

We computed a mediation model in *lavaan* with occurrence of tears as the predictor, perceived honesty as the mediator, and social support intentions as the outcome. The model was tested separately for different levels of target gender and situational context, again using 10,000 bootstrap repetitions for the confidence intervals of the indirect effects. An overview of indirect effects is provided in Table 4, and a detailed overview is in Supplementary Table S25. We observed a significant positive indirect effect for manipulative contexts, male targets, and the overall effect (with no moderator). For manipulative contexts and male targets, tears increased perceptions of honesty, which in turn positively correlated with support intentions. The effect was smaller for non-manipulative contexts and female targets and not significant.

**H4. Moderation by the Dark Triad:**   Next, we performed a model with perceived honesty as the outcome and occurrence of tears, situational context, and all three Dark Triad indicators (centered) as predictors (Supplementary Table S28). We observed a significant main effect of situational context ($d = −.41$ [−.55, −.027], $p < .001$) and a significant interaction effect between the occurrence of tears and psychopathy ($\beta = −.20$ [−.37, −.03], $p = .022$). Individuals high in psychopathy rated tearful targets as less honest, similar to findings from Study 1 (Fig 5). We observed similar effects when running models for each Dark Triad predictor separately (Supplementary Table S29).

**H5. Moderation by perceived appropriateness:**   Finally, we repeated the model from H1 with perceived appropriateness instead of honesty (Supplementary Table S30). We only observed a significant main effect of situational context with expressions in manipulative contexts perceived as less appropriate ($d = −.43$ [−.63, −.22], $p < .001$). Exploring the three-way interaction, we observed a tendency for tears to be perceived as more appropriate in manipulative contexts for male targets ($d = .13$ [−.08, .32]) compared to female targets ($d = −.02$ [−.20, .16]), providing limited evidence for a similar direction as observed in Study 1 (Supplementary Fig S9).

## Discussion

Using a different design and warmth manipulation based on the target gender, we partially replicated the effects we observed in Study 1. Most importantly, similar to Study 1, we found that the overall effect of tears on perceived honesty was very small, yet its strength varied depending on contextual factors.

Similar to Study 1, Study 2 revealed a pattern suggesting that the social effects of tears might be slightly more beneficial for male (low-warmth) than female (high-warmth) targets. Specifically, tears increased honesty ratings somewhat more for men than women, but this difference did not reach statistical significance. Similarly, we observed a tendency

suggesting that tears increased perceived manipulativeness more for female than male targets, but again, this effect was not significant. At the same time, we replicated the mediation effect, indicating that tears increased support intentions via perceived honesty more strongly for male than female targets. The direct positive effect of tears on support intentions was also more pronounced for male than female targets, especially when they were depicted in manipulative contexts. Overall, these effects lend additional credence to the idea that men/low-warmth targets might benefit more from a warmth-signaling expressive behavior (i.e., shedding tears) than women/high-warmth targets.

Study 2 provided no support for the role of the target manipulativeness ratings and expression authenticity ratings for perceived honesty because, surprisingly, the presence of tears did not affect these ratings. Moreover, in contrast to Study 2 and other studies [2,3,31,75], we found no support for the role of tears in shaping perceived appropriateness ratings. However, in line with Study 1, we observed that in manipulative contexts, male tears were perceived as more appropriate than female tears, which again suggests that male tears may lead to more socially beneficial effects.

Finally, when testing the role of the Dark Triad traits for the perception of tearful vs. non-tearful targets, we replicated the finding that individuals high in psychopathy perceived targets shedding tears as less honest than targets with no tears, thereby corroborating the effect found in Study 1. Yet, contrary to Study 2, we found no similar support for the moderating role of Machiavellianism.

Overall, Study 2 provided limited evidence for the effects we already observed in Study 1. Most importantly, the effects of warmth/gender we observed this time, despite converging with the findings of Study 1, were weaker, and one reason for that was the warmth manipulation we used. Specifically, although we selected female targets rated highest on warmth and male targets rated lowest on warmth based on a previous study [15], our manipulation was only half as strong as the face warmth manipulation we employed in Study 1. At the same time, using more ecologically valid pictures of tearful and non-tearful targets resulted in showing clear evidence for the direct positive impact of tears on social support intentions (i.e., the effect reported in many previous studies [3,15,16], which we failed to replicate in Study 1). We should also note that despite a large sample (~1,600 participants), Study 2 might not have been sufficiently powered to detect very small interaction effects, which might also explain why the results of Study 1 were not fully replicated.

## Meta-analytical integration

To provide a synthesized overview and address problems of individual study power, we performed random effects meta-analyses across the effects obtained in Studies 1 and 2 (for each country/sample separately) using the *metafor* [76] and *metaSEM* [77] packages. An overview of meta-analytical main effects is provided in Table 5, and an overview of meta-analytical mediation effects is provided in Table 4.

## General discussion

Popular media and common wisdom suggest that the tendency to perceive tears as sincere social signals may be considerably reduced under specific circumstances. Here, we addressed this possibility by testing whether contextual factors may increase the perceived manipulativeness of a tearful target and decrease the perceived authenticity of their tears, thereby reducing the target's perceived honesty and, in turn, making the observers less willing to provide help. Our findings supported the notion that context matters for the perception of tears and their social effects. More specifically, when the effect of tears on honesty ratings was tested in previous studies [11] or the current Preliminary Study ($d = .28$) in which no manipulation with the situational context manipulativeness or target characteristics was introduced, the effect of tears on perceived honesty was considerably stronger than in Studies 1 and 2 ($d = -.02$) in which targets explicitly represented low- vs. high-warmth individuals and were depicted in specifically manipulative vs. non-manipulative situations contexts. Overall, this suggests that tears influence honesty only under specific conditions, replicating previous findings on the highly context-dependent effects of tears on competence ratings [14,31,32]. At the same time, the role of many contextual factors turned out to be more complex and much weaker than we initially predicted.

**Table 5. Overview of meta-analytical effects for perceived honesty, perceived manipulativeness, perceived expression authenticity, and support intentions across studies 1 and 2.**

| Variable | Moderator | Moderator Level | Estimate (Cohen's d) | 95% CI | M | Q | I | R2 |
|---|---|---|---|---|---|---|---|---|
| **Main Effects** | | | | | | | | |
| Honesty | | | −.02 | [−.10, .07] | | (4) = 18.59, p < .001 | 79.27% | |
| | Face Warmth | Low Warmth | .13 | [.04, .22] | (1) = 17.70, p < .001 | (6) = 3.09, p = .798 | 0.00% | 100% |
| | | High Warmth | −.14 | [−.23, −.05] | | | | |
| | Target Gender | Female | −.03 | [−.11, .05] | (1) = 4.34, p = .037 | (8) = 8.87, p = .353 | 17.78% | 61.47% |
| | | Male | .09 | [.00, .18] | | | | |
| | Situational Context | Non-Manipulative | .07 | [−.01, .15] | (1) = 1.60, p = .206 | (8) =10.34, p = .242 | 27.03% | 0.00% |
| | | Manipulative | −.02 | [−.13, .09] | | | | |
| | Target Gender * Situational Context | Female / Non-Manipulative | .02 | [−.09, .13] | (1) = .11, p = .738 | (18) = 22.28, p = .219 | 18.31% | 0.00% |
| | | Male / Non-Manipulative | .12 | [.01, .23] | | | | |
| | | Female / Manipulative | −.09 | [−.23, .05] | | | | |
| | | Male / Manipulative | .06 | [−.06, .17] | | | | |
| Manipulative-ness | | | .13 | [.04, .22] | | (4) = 10.74, p < .029 | 62.72% | |
| | Face Warmth | Low Warmth | .16 | [.03, .30] | (1) = .02, p = .891 | (6) = 8.51, p = .203 | 25.77% | 0.00% |
| | | High Warmth | .16 | [.06, .25] | | | | |
| | Target Gender | Female | .16 | [.08, .24] | (1) = 1.14, p = .285 | (8) = 11.87, p = .157 | 32.11% | 3.20% |
| | | Male | .09 | [−.01, .20] | | | | |
| | Situational Context | Non-Manipulative | .07 | [.00, .15] | (1) = 2.17, p = .141 | (8) = 12.81, p = .119 | 37.17% | 26.09% |
| | | Manipulative | .18 | [.05, .31] | | | | |
| | Target Gender * Situational Context | Female / Non-Manipulative | .13 | [.02, .24] | (1) = 1.44, p = .229 | (18) = 18.90, p = .398 | 1.76% | 55.11% |
| | | Male / Non-Manipulative | .01 | [−.10, .12] | | | | |
| | | Female / Manipulative | .18 | [.06, .31] | | | | |
| | | Male / Manipulative | .17 | [.04, .31] | | | | |
| Authenticity | | | −.03 | [−.11, .05] | | (4) = 8.65, p = .071 | 53.23% | |
| | Face Warmth | Low Warmth | .01 | [−.10, .11] | (1) = 4.33, p = .038 | (6) = 5.99, p = .424 | .06% | 99.88% |
| | | High Warmth | −.13 | [−.22, −.04] | | | | |
| | Target Gender | Female | −.04 | [−.12, .04] | (1) = .22, p = .641 | (8) = 13.66 p = .091 | 41.99% | 0.00% |
| | | Male | −.01 | [−.14, .11] | | | | |
| | Situational Context | Non-Manipulative | .04 | [−.04, .11] | (1) = 3.43, p = .064 | (8) = 12.84 p = .117 | 39.70% | 31.67% |
| | | Manipulative | −.10 | [−.23, .03] | | | | |
| | Target Gender * Situational Context | Female / Non-Manipulative | .06 | [−.05, .17] | (1) = 2.54, p = .111 | (18) = 23.76 p = .163 | 26.30% | 11.51% |
| | | Male / Non-Manipulative | .01 | [−.11, .13] | | | | |
| | | Female / Manipulative | −.14 | [−.27, −.01] | | | | |
| | | Male / Manipulative | −.05 | [−.19, .10] | | | | |
| Support Intentions | | | .09 | [.01, .16] | | (4) = 7.22, p < .125 | 45.22% | |
| | Face Warmth | Low Warmth | .20 | [.11, .29] | (1) = 19.90, p < .001 | (6) = 4.71, p = .581 | 0.00% | 100% |
| | | High Warmth | −.09 | [−.18, .00] | | | | |
| | Target Gender | Female | .00 | [−.10, .11] | (1) = 5.30, p = .021 | (8) = 11.72, p = .164 | 33.74% | 56.51% |
| | | Male | .17 | [.09, .25] | | | | |
| | Situational Context | Non-Manipulative | .12 | [.03, .21] | (1) = 1.09, p = .296 | (8) = 7.83, p = .449 | 7.37% | 0.00% |
| | | Manipulative | .06 | [−.01, .14] | | | | |
| | Target Gender * Situational Context | Female / Non-Manipulative | .08 | [−.06, .22] | (1) = .009, p = .922 | (18) = 23.17, p = .184 | 24.93% | 0.00% |
| | | Male / Non-Manipulative | .16 | [.05, .27] | | | | |
| | | Female / Manipulative | −.04 | [−.15, .06] | | | | |
| | | Male / Manipulative | .16 | [.06, .27] | | | | |

## Target characteristics: Warmth and gender

Despite our initial hypotheses, Study 1 revealed that the effect of face warmth on perceived honesty was the opposite of what we expected. Even though high-warmth faces were generally perceived as more honest than low-warmth faces, tears reduced this difference by slightly increasing perceived honesty for low-warmth faces and slightly decreasing it for high-warmth faces. The effects we observed for perceived target manipulativeness and perceived expression authenticity also converged with the idea that tears led to more beneficial effects when participants rated targets low in warmth.

Notably, a similar pattern emerged when we tested the moderating role of target gender. Study 1 revealed that the effect of tears on perceived honesty was positive for male targets but negative for female targets, which further affected support intentions: Tears increased support intentions in response to a tearful man but not in response to a tearful woman. Driven by these findings, in Study 2, we combined target warmth and target gender manipulation, using pictures of women who were high in warmth and men who were low in warmth. By doing so, we partially replicated the findings of Study 1. Specifically, although we did not find evidence that the effect of tears on honesty ratings was reversed for female targets, we observed a slight tendency suggesting that tears increased perceived honesty more strongly for men than women. Further and even more importantly, the effect of tears on support intentions – both direct and indirect (through perceived honesty) – was more pronounced for men than women, supporting the notion that tears are more beneficial for men and more detrimental for women, and replicating a recently published study on manipulative crying which showed that female criers may face more backlash than male criers [56].

On the one hand, these results seem counterintuitive because warmth and honesty are positively correlated [37]. Accordingly, women/high-warmth targets should be perceived as less likely to shed crocodile tears. On the other hand, it is possible that warm targets and women benefited less from shedding tears because they already possessed the characteristics typically associated with tearful individuals. Put differently, given that tears are consistently associated with high warmth (for a review, see [14]), adding tears to already warm faces might have exaggerated the social information conveyed by these faces, thereby increasing participants' suspiciousness of the targets' sincere intentions, and reducing the beneficial social effects of tears. This explanation is supported by a widespread belief that being excessively friendly is superficial and calculated and, thus, is a commonly practiced manipulation tactic [78]. In line with this belief, we observed that the effects of target characteristics were stronger when we manipulated objective facial features indicative of warmth in Study 1 (thereby exaggerating the difference between high- and low-warmth targets) than when we used a more subtle gender-based warmth manipulation in Study 2.

Another possible explanation may be related to the fact that both warmth and female gender are positively associated with high emotionality [79], and hence, tears shed by women and warm individuals might be seen as a manifestation of their routine expressive behavior, idiomatically referred to as "the boy cried wolf" (a phrase used to describe the tendency to raise the alarm over nothing). Accordingly, tears shed by women and warm people may be attributed to trivial reasons and, in consequence, damage their credibility in the eyes of others. By analogy, men's and low-warmth individuals' tears, being less common and less expected, might be attributed to more critical and thus genuine reasons and thus lead to more socially favorable effects [72]. Future research is thus needed to test whether differences in the perceived honesty of tearful individuals can be explained by observers' interpretation of the reasons behind shedding tears.

## The manipulativeness of situational context and perceived expression appropriateness

Our research also confirmed that the perceptions of honesty may depend on situational context. Interestingly, as demonstrated by the Preliminary Study, the effect of situational settings on perceived honesty emerged even when no explicit information about context manipulativeness was provided, as participants evaluated the targets only based on whether the context was emotional or not. In general, tears shed for emotional (positive or negative) reasons were perceived as a signal of honesty, which was not the case for tears shed for no reason (in neutral situations). This result not only highlights

the already-mentioned importance of reasons for shedding tears but also suggests that the targets were perceived as more honest when their tears matched the context in which they appeared, thereby pointing to the role of perceived expression appropriateness (see also [3]).

Studies 1 and 2 further addressed the role of situational context and perceived expression appropriateness. First, both studies confirmed that manipulative situational contexts (in which targets were portrayed as exerting social influence), compared to non-manipulative situational contexts (in which no information about social influence was provided), were associated with lower perceived target honesty, higher perceived target manipulativeness, and lower perceived expression authenticity, and these effects were further modulated by tears. Yet, the results across both studies were not entirely consistent. Study 1 revealed a tendency suggesting that tears reduced honesty ratings and increased manipulativeness ratings slightly more when targets were depicted in manipulative than non-manipulative situations. Moreover, in line with our reasoning, expressions of targets shedding tears in manipulative situations, in contrast to non-manipulative situations, were evaluated as less authentic than expressions of targets not shedding tears. Study 2, however, found no significant interactions between the occurrence of tears and situational context.

Second, both studies indicated that the expressions of the targets presented in manipulative situations were evaluated as less appropriate than expressions of targets presented in non-manipulative situations, resonating with studies showing that the perception of emotional expressions depends on the situation in which these expressions occur [80,81]. Study 1 also showed that the perception of expression appropriateness was modulated by the occurrence of tears, situational context, and face warmth. More specifically, in manipulative contexts, expressions of high-warmth targets were evaluated as more appropriate when presented with no tears, while for low-warmth targets, we observed the opposite pattern (i.e., higher appropriateness ratings when they shed tears). Testing the corresponding three-way interaction in Study 2 revealed no significant effect, but an identical tendency emerged, lending additional yet limited support for the interplay among tears, situational context, and target warmth/gender.

Together, these results partially support the role of context and perceived expressed appropriateness in evaluating honesty-related target characteristics, but we observed many inconsistencies across both studies. One reason might be that our situational context manipulation, even though successful, was not strong enough. This was most probably related to the fact that we wanted to avoid explicitly morally wrong or good situations, which eliminated situational contexts that were highly manipulative. Therefore, future research is needed to examine whether tears shed in contexts that are more manipulative than the ones we presented participants with are more strongly related to perceptions of the target's manipulativeness and honesty, as well as expression authenticity.

### The effect of tears on social support

In line with the social glue model of emotional tears [14], all three studies confirmed that the effect of tears on the willingness to support a tearful target could be explained by perceived honesty. Studies 1 and 2 also showed that this mediation effect may be moderated by target warmth/gender. Specifically, in Study 1, we found that the indirect effect of tears via perceived honesty was positive for low-warmth targets and negative for high-warmth targets. Study 2 partially replicated this finding, showing that tears indirectly increased support intentions for male (low-warmth) targets, while for female targets, this effect was non-significant.

Overall, these findings mirror the already-discussed effects of tears on observers' inferences, showing that the impact of tears on support intentions was more beneficial for low-warmth than high-warmth targets. On the one hand, these results may seem surprising given that previous studies have consistently shown that the positive effect of tears on observers' willingness to help a tearful target can be attributed to the increase in perceived warmth (for a review, see [14]). However, we should note that none of these studies manipulated the target's warmth directly. Instead, participants were presented with targets characterized by neutral characteristics, and only subjective perceptions of warmth were measured. Here, by employing pictures of targets varying in objective characteristics indicative of warmth (i.e., facial features

and gender), we were able to show that the social effects of tears (in particular, their impact on support intentions) may become more positive when a warmth-signaling expression such as tears appear on faces low in warmth.

### The observer's Dark Triad personality traits

Our analyses also showed that participants scoring high on psychopathy (Studies 1 and 2) and Machiavellianism (only Study 1) evaluated targets shedding tears as less honest than targets with no tears. These results suggest that people scoring high on psychopathy are more likely to question the authenticity of others' expressed distress [22], which converges with the increasingly popular notion that the perception of emotional expressions depends on a broad category of contextual factors, and observers' characteristics are an essential element of this category [20,81].

### Culture

The CCT re-analysis suggested that the tendency to associate tears with honesty may vary across countries. Study 1 supported this possibility, showing that tears are linked to perceived manipulativeness more strongly for participants representing low-trust countries (South Africa and Poland) than high-trust countries (Norway and Canada). At the same time, the lack of similar effects on perceived honesty, expression authenticity, and support intentions suggests that the impact of culture is limited. Hence, future studies are needed to test which inferences are subject to the influence of cultural factors and which are not.

### Limitations and future directions

The present study is not free of limitations. First, the effects we observed were small, and the results were not entirely consistent across the studies. Most importantly, these inconsistencies seem to be driven, in particular, by the warmth manipulation we used. Hence, when planning future studies on the effects of warmth/gender on the social role of tears, researchers must carefully decide how to introduce this factor. At the same time, it is possible that the fact that the effects of tears varied depending on specific target features may explain large inconsistencies in previous studies testing differences in the overall perception of tearful men and women (for a review, see [3]). Given the results of our current research, it seems likely that these inconsistencies may be, at least to some extent, stimuli-dependent (for instance, some databases may include male and female faces that are more varied in terms of facial warmth than others; see [3] where male and female targets on average did not differ significantly on perceived warmth).

Second, although using static pictures of strangers accompanied by situational vignettes is a common practice in research on the social effects of tears (for a review, see [11]), the ecological validity of this method is somewhat limited [23,82]. In reality, people often shed tears in the presence of their close ones [82,83], and their expression is naturally embedded in the context (while using vignettes requires participants to combine two sources of information). Moreover, in all three studies, pictures of tearful individuals were created by adding tears digitally. Even though this technique is common in experimental research on tears, and its effects are comparable to using pictures with genuine tears (for a review, see [84]), it is possible that such tears may look unnatural (and hence less authentic) due to several characteristics that might be difficult to control (e.g., their unnatural flow or reflectivity). Future research would thus benefit from employing more ecologically valid methods.

Finally, when testing the mechanisms behind the effects of tears on honesty ratings or support intentions, we employed a measurement-of-mediation design. Therefore, our conclusions regarding causal processes need to be treated with caution [85], and future studies would benefit from manipulating the mediators. The results of the present study clearly show how critical this need is by indicating that the effects of manipulated target warmth were contrary to what we already know from studies where only perceived warmth was measured.

## Conclusions

Using large datasets and employing rich sets of stimuli, we found that the effects of tears on perceptions of honesty cannot be studied outside of the context in which tears appear because these effects are driven by many contextual factors. Most importantly, we found that shedding tears may be more socially beneficial when done in non-manipulative social contexts and by those who are not expected to engage in such a warmth-signaling expressive behavior (i.e., low-warmth targets and men).

## Supporting information

**S1 Note. Preliminary Study: Re-Analysis of the CCT Project (Table S1 & Figure S1).**
(DOCX)

**S2 Note. Country Selection (Figure S2).**
(DOCX)

**S3 Note. Vignette Ratings (Tables S2-S3 & Figure S3).**
(DOCX)

**S4 Note. Study 1 Additional Analyses (Tables S4-S21 & Figures S4-S7).**
(DOCX)

**S5 Note. Study 2 Additional Analyses (Tables S22-S30 & Figures S8-S9).**
(DOCX)

## Author contributions

**Conceptualization:** Monika Wróbel, Janis H. Zickfeld, Paweł Ciesielski.

**Funding acquisition:** Monika Wróbel.

**Investigation:** Monika Wróbel, Janis H. Zickfeld, Paweł Ciesielski.

**Methodology:** Monika Wróbel, Janis H. Zickfeld, Paweł Ciesielski.

**Resources:** Monika Wróbel, Janis H. Zickfeld, Paweł Ciesielski.

**Validation:** Monika Wróbel, Janis H. Zickfeld, Paweł Ciesielski.

**Writing – original draft:** Monika Wróbel, Janis H. Zickfeld.

**Writing – review & editing:** Monika Wróbel, Janis H. Zickfeld.

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
