## [Decision Letter · Decision Letter 0]

PONE-D-25-04776The honesty behind tears: Situational, individual, and cultural influences of perceiving emotional tears as sincerePLOS ONE

Dear Dr. Wróbel,

Thank you for submitting your manuscript to PLOS ONE. After careful consideration, we feel that it has merit but does not fully meet PLOS ONE’s publication criteria as it currently stands. Therefore, we invite you to submit a revised version of the manuscript that addresses the points raised during the review process. I agree with both reviewers that the manuscript needs a more focused literature review and a clear description of the methods used. Thus, I invite the authors to address all the raised issues and resubmit for acceptance. Please submit your revised manuscript by May 29 2025 11:59PM. If you will need more time than this to complete your revisions, please reply to this message or contact the journal office at plosone@plos.org . Please include the following items when submitting your revised manuscript:

We look forward to receiving your revised manuscript.

Kind regards,

Vilfredo De Pascalis

Academic Editor

PLOS ONE

3. We note that Figure 3 includes an image of participant in the study.

Additional Editor Comments:

I agree with both reviewers that the manuscript needs a more focused literature review and a clear description of the methods used. Thus, I invite the authors to address all the raised issues and resubmit for acceptance.

Reviewers' comments:

Reviewer's Responses to Questions

**Comments to the Author**

1. Is the manuscript technically sound, and do the data support the conclusions?

Reviewer #1: Yes

Reviewer #2: Partly

2. Has the statistical analysis been performed appropriately and rigorously? 

Reviewer #1: Yes

Reviewer #2: Yes

3. Have the authors made all data underlying the findings in their manuscript fully available?

Reviewer #1: Yes

Reviewer #2: Yes

4. Is the manuscript presented in an intelligible fashion and written in standard English?

Reviewer #1: Yes

Reviewer #2: Yes

5. Review Comments to the Author

Reviewer #1: It analyses the situational context, and does not include the cultural context, as there are several samples from different cultures (South Africa, Canada, Poland, Norway, United Kingdom), methodologically it does not specify the reason for the selection of the sample from these countries, and on the other hand in the results and discussion section it does not expand on the existing cultural differences.

The concept of ‘warmth in the face’ is not sufficiently defined for further measurement and comparison of variables.

The situational variable referred to in the study should be made more specific.

Reviewer #2: 1. Strengthening the theoretical and literature foundation:

Please provide more theoretical basis for "emotional tears as honesty signals", especially the relevant literature on evolutionary psychology, honest behavior and non-verbal signal theory, to strengthen the theoretical context of this study.

The tension between "crocodile tears" and honest signals deserves further exploration. It is recommended to clearly distinguish the different functions that tears may play in social situations (such as seeking support, manipulation, trust building, etc.).

2. The methodology needs to be explained more clearly and refined:

Please clearly explain what are "posed" and "genuine" tear images, and provide specific examples or verification methods to explain the standards for distinguishing between real and fake tears.

Describe the design details of the manipulative vs. non-manipulative situations and whether the manipulation test method is effective

6. PLOS authors have the option to publish the peer review history of their article (what does this mean? ). If published, this will include your full peer review and any attached files.

**Do you want your identity to be public for this peer review?** For information about this choice, including consent withdrawal, please see our Privacy Policy .

Reviewer #1: **Yes: ** JOSE LUIS GIL BERMEJO

Reviewer #2: No

---

## [Editor Report · Decision Letter 1]

The honesty behind tears: Situational, individual, and cultural influences of perceiving emotional tears as sincere

PONE-D-25-04776R1

Dear Dr. Wróbel,

We’re pleased to inform you that your manuscript has been judged scientifically suitable for publication and will be formally accepted for publication once it meets all outstanding technical requirements.

Kind regards,

Vilfredo De Pascalis

Academic Editor

PLOS ONE

Additional Editor Comments (optional):

I testify that all the suggested changes have been addressed with revision. Thus the current version of the manuscript can be accepted for publication.
---

## [Editor Report · Acceptance letter]

PONE-D-25-04776R1

PLOS ONE

Dear Dr. Wróbel,

I'm pleased to inform you that your manuscript has been deemed suitable for publication in PLOS ONE. Congratulations! Your manuscript is now being handed over to our production team.

Kind regards,

on behalf of

Prof. Vilfredo De Pascalis

Academic Editor

PLOS ONE